# Switch-like control of helicase processivity by single-stranded DNA binding protein

**Barbara Stekas[1], Steve Yeo[2], Alice Troitskaia[2], Masayoshi Honda[3], Sei Sho[3], Maria Spies[3], Yann R Chemla[1,2,4]\***

[1]Department of Physics, University of Illinois, Urbana-Champaign, Urbana, United States; [2]Center for Biophysics and Quantitative Biology, University of Illinois, Urbana-Champaign, Urbana, United States; [3]Department of Biochemistry, Carver College of Medicine, University of Iowa, Iowa City, United States; [4]Center for the Physics of Living Cells, University of Illinois, Urbana-Champaign, Urbana, United States

**Abstract** Helicases utilize nucleotide triphosphate (NTP) hydrolysis to translocate along single-stranded nucleic acids (NA) and unwind the duplex. In the cell, helicases function in the context of other NA-associated proteins such as single-stranded DNA binding proteins. Such encounters regulate helicase function, although the underlying mechanisms remain largely unknown. *Ferroplasma acidarmanus* xeroderma pigmentosum group D (XPD) helicase serves as a model for understanding the molecular mechanisms of superfamily 2B helicases, and its activity is enhanced by the cognate single-stranded DNA binding protein replication protein A 2 (RPA2). Here, optical trap measurements of the unwinding activity of a single XPD helicase in the presence of RPA2 reveal a mechanism in which XPD interconverts between two states with different processivities and transient RPA2 interactions stabilize the more processive state, activating a latent 'processivity switch' in XPD. A point mutation at a regulatory DNA binding site on XPD similarly activates this switch. These findings provide new insights on mechanisms of helicase regulation by accessory proteins.

**\*For correspondence:**
ychemla@illinois.edu

## Introduction

Helicases are molecular machines that use the energy of NTP hydrolysis to separate the strands of nucleic acid (NA) duplexes. A large class of helicases unwind by translocating directionally along a single NA strand and can utilize this translocation activity to displace NA-bound proteins (*Delagoutte and von Hippel, 2002*; *Delagoutte and von Hippel, 2003*; *Singleton et al., 2007*; *Lohman et al., 2008*). In the cell, helicases are involved in many essential genome maintenance processes, including replication, recombination, and repair (*McGlynn, 2013*; *Daley et al., 2013*; *Kuper and Kisker, 2013*). In a number of instances, the same helicase is known to carry out several of these disparate functions (*Lohman et al., 2008*; *Wu and Spies, 2013*; *Beyer et al., 2013*). Thus, helicase activity must be tightly regulated not only to prevent indiscriminate unwinding of duplex DNA in the cell, which could lead to genome instability, but also to define the context-dependent role of the helicase. How this regulation occurs is often unclear, but growing evidence points to interactions with protein partners as one regulatory mechanism (*Lohman et al., 2008*; *Wu and Spies, 2013*; *Beyer et al., 2013*).

Xeroderma pigmentosum group D (XPD) protein is a 5′ to 3′ DNA helicase that serves as a model for understanding members of the structural superfamily 2B of helicases, a group that includes yeast Rad3 and human FANCJ, RTEL, and CHLR1 (*Fairman-Williams et al., 2010*; *Byrd, 2012*; *White and Dillingham, 2012*; *Beyer et al., 2013*). Human XPD is part of transcription factor IIH and plays a vital role in nucleotide excision repair (*Egly and Coin, 2011*; *Fuss and Tainer, 2011*; *Kuper et al., 2014*;

*Houten et al., 2016*). It has also been shown to participate in chromosome segregation (*Ito et al., 2010*) and in the cell's defense against retroviral infection (*Yoder et al., 2006*). Structurally, XPD consists of a conserved motor core (*Figure 1a*)—helicase domains 1 and 2 (HD1 and HD2)—that couples ATP binding and hydrolysis to translocation on ssDNA, an FeS cluster-containing domain, and a unique ARCH domain that encircles the translocating strand (*Fan et al., 2008*; *Liu et al., 2008*; *Wolski et al., 2008*; *Kuper et al., 2012*; *Spies, 2014*).

As it functions on ssDNA, XPD is expected to encounter other proteins such as replication protein A (RPA), which binds single-stranded DNA non-sequence specifically. Single-stranded DNA binding proteins function as protectors of ssDNA against nucleolytic damage and play important regulatory roles (*Shereda et al., 2008*; *Caldwell and Spies, 2020*). Due to their ubiquity, frequent interactions with other DNA-bound proteins are likely (*Spies and Ha, 2010*). Single-stranded binding proteins are known to stimulate the DNA unwinding activity of SF2 helicases, although the mechanisms of enhancement remain elusive (*Harmon and Kowalczykowski, 2001*; *Rajagopal and Patel, 2008*; *Cadman and McGlynn, 2004*; *Cui et al., 2004*; *Gupta et al., 2007*; *Pugh et al., 2008*;

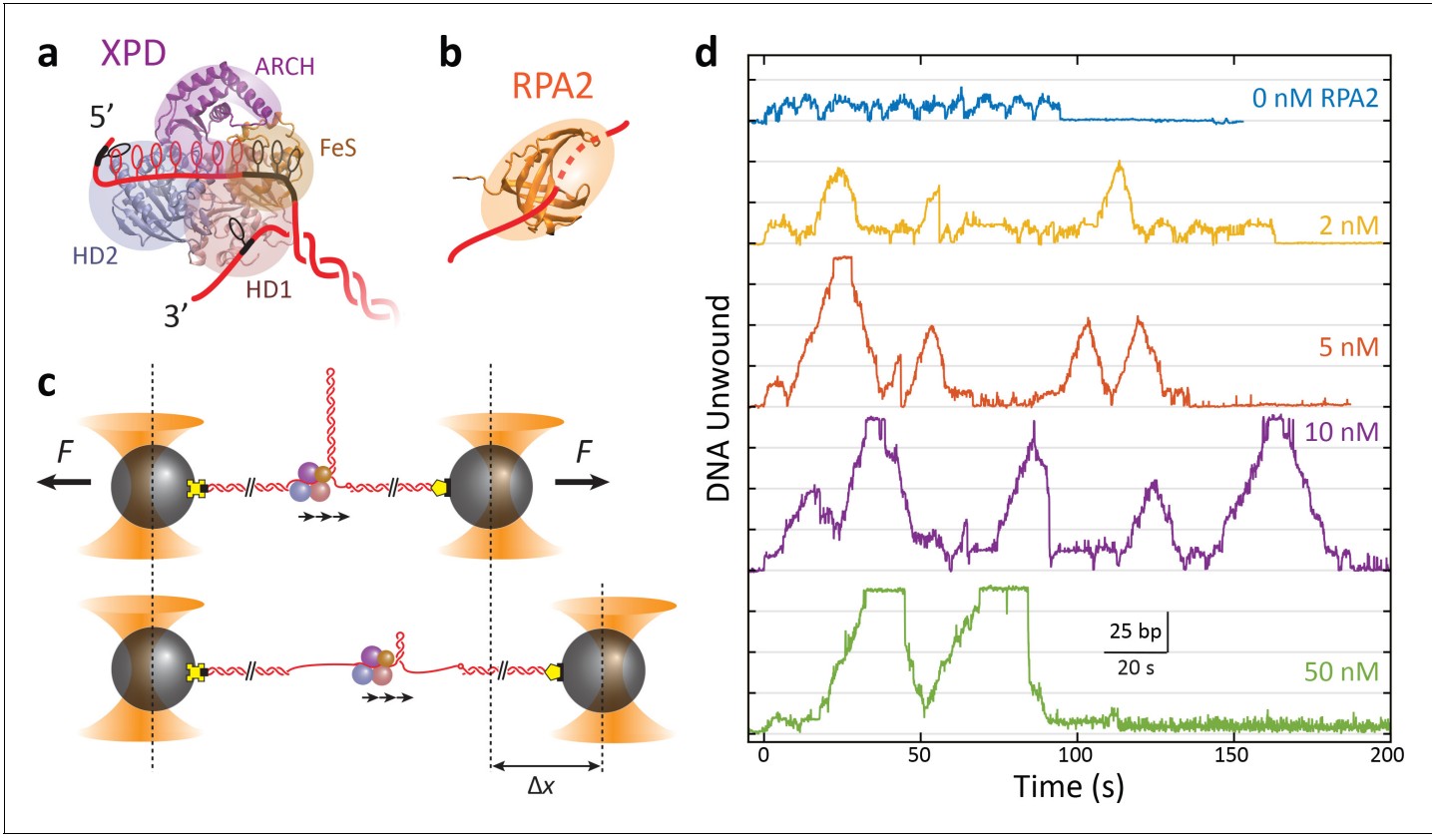

**Figure 1.** RPA2 increases xeroderma pigmentosum group D (XPD) helicase processivity. (**a**) Schematic of *Fac*XPD based on *Sulfolobus acidocaldarius* XPD structure (PDB 3CRV). XPD is composed of four domains: helicase domains 1 and 2 (HD1, pink; HD2, blue), which form the motor core, and the ARCH domain (purple) and FeS cluster (brown). A model of the DNA fork (red) shows ~10 nt bound to the motor core and secondary contacts (black) to each strand of the fork. (**b**) Schematic of *Fac*RPA2 based on partial crystal structure of *Methanococcus maripaludis* replication protein A (RPA) (PDB 2K5V). RPA2 consists of a single OB-fold that binds to ssDNA in the C-shaped cavity. (**c**) Single-molecule hairpin unwinding assay. A DNA hairpin consisting of an 89 bp stem and $(dT)_4$ loop is tethered between trapped beads by biotin–streptavidin (yellow cross) and digoxigenin-antibody linkages (pentagon) and held at a constant force. A 10 nt poly-dT site at the 5′ end of the hairpin allows one XPD molecule to load, and an abasic site at the 3′ end prevents XPD unwinding past the hairpin. Unwinding of the hairpin by XPD increases the end-to-end extension of the construct by an amount, Δ*x*, proportional to the number of base pairs unwound. Arrows indicate the 5′ to 3′ direction of XPD translocation along ssDNA. (**d**) Representative traces of a single molecule of XPD unwinding in the presence of varying concentrations of RPA2 (0–50 nM) at constant force (*F* = 12 pN). ATP and RPA2 are added at *t* = 0 s. XPD processivity increases with RPA2 concentration.

The online version of this article includes the following figure supplement(s) for figure 1:

**Figure supplement 1.** DNA hairpin construct.
**Figure supplement 2.** Laminar flow chamber.

*Bétous et al., 2013*). In the present work, we investigate the effect of single-stranded DNA binding proteins on helicase unwinding activity using XPD from *Ferroplasma acidarmanus* as a model system due to the availability of biochemical (*Pugh et al., 2008*) and single-molecule kinetic data (*Honda et al., 2009*; *Qi et al., 2013*; *Ghoneim and Spies, 2014*), as well as structural information from homologs (*Liu et al., 2008*; *Wolski et al., 2008*; *Kuper et al., 2012*; *Wolski et al., 2010*). Prior work (*Pugh et al., 2008*) has shown that *Fac*XPD unwinding is enhanced greatly by one type of cognate ssDNA binding protein in *F. acidarmanus*, RPA2, more so than a second cognate protein, RPA1, or by heterologous proteins. *Fac*RPA2 has a simple, single-domain architecture with one OB-fold that occludes ~4 nt of ssDNA (*Figure 1b*; *Pugh et al., 2008*). Single-molecule experiments demonstrated that XPD can occupy the same DNA strand as RPA2 and bypass RPA2-bound ssDNA (*Honda et al., 2009*). Crucially, no direct, specific interactions between RPA2 and XPD in solution have been observed (*Pugh et al., 2008*).

How an ssDNA binding protein with no known protein–protein contacts to the helicase stimulates helicase-mediated DNA unwinding activity remains an open question and makes *Fac*XPD and *Fac*RPA2 an intriguing system for studying the stimulatory effects of ssDNA binding proteins on helicases. Mechanisms for enhancement of unwinding activity can loosely be placed in three categories (*Pugh et al., 2008*). First, an ssDNA binding protein could destabilize the DNA duplex at the ssDNA–dsDNA junction ahead of the helicase, facilitating its motion forward. Second, it could enhance unwinding by sequestering ssDNA and providing a physical barrier to helicase backsliding, that is, rectifying helicase forward motion. Lastly, it could activate the helicase for processive unwinding through direct interactions with the helicase-DNA complex.

Here, we use a single-molecule optical trap assay to observe individual molecules of XPD unwinding DNA and to analyze the effects of RPA2 on their unwinding activity. While previous reports have shown that RPA2 is able to enhance XPD activity (*Pugh et al., 2008*; *Honda et al., 2009*), they have not defined what aspects of activity are enhanced or provided a mechanism. Our results show that RPA2 primarily increases XPD processivity or the maximum number of base pairs unwound. XPD exhibits repeated attempts to unwind duplex DNA, and RPA2 increases the frequency of attempts that have high processivity. While RPA2 can transiently destabilize duplex DNA, this does not promote XPD unwinding. Instead, data point to a mechanism in which XPD possesses a latent processivity 'switch', which is activated specifically by RPA2. We propose that an integral component of this processivity switch is the regulatory interaction between DNA and a secondary binding site on XPD, as supported by a point mutation at this site that enhances XPD processivity similarly to RPA2. Our measurements shed new light on the mechanisms by which accessory proteins such as single-stranded DNA binding proteins can enhance helicase activity.

## Results

### XPD processivity increases in the presence of RPA2

As shown in *Figure 1c*, we monitored the activity of a single XPD in the absence and presence of RPA2 in solution from the unwinding of an 89 bp DNA hairpin (*Figure 1—figure supplement 1*) tethered between two optically trapped beads and stretched under constant force, as described previously (*Qi et al., 2013*). Measurements were carried out over a range of forces (7–12 pN) at which single-molecule XPD activity was previously reported (*Qi et al., 2013*) and below that required to mechanically unfold the hairpin (15 pN; *Figure 1—figure supplement 1*). Data were collected at a rate of 89 Hz. Unwinding of the hairpin was detected from the increase in the end-to-end extension of the DNA tether as each broken base pair released two nucleotides (see Materials and methods). A 10 dT ssDNA site—approximately equal to the footprint of XPD (*Qi et al., 2013*; *Kokic et al., 2019*)—at the 5′ end of the hairpin served as a binding site for a single XPD molecule, and an abasic site positioned at the 3′ end of the hairpin prevented XPD from unwinding the long (1.5 kb) dsDNA handle used to separate the hairpin from the trapped beads (*Qi et al., 2013*; *Figure 1—figure supplement 1*). To control the loading of XPD and RPA2 onto the hairpin, we used a custom flow chamber consisting of parallel laminar-flow streams containing different buffers, as described previously (*Whitley et al., 2017*) (see Materials and methods; *Figure 1—figure supplement 2a*). The hairpin was placed in a protein 'loading' stream containing XPD but no ATP for ~1 min., which allowed a single XPD to bind to the 10 dT loading site without unwinding the

DNA. Unwinding was initiated by moving the hairpin and bound XPD into the 'unwinding' stream containing saturating ATP (500 µM), varying concentrations of RPA2 (0–50 nM), and no additional XPD in solution (*Figure 1—figure supplement 2b, c*). This procedure ensured that only a single XPD was loaded at one time, and that the unwinding observed resulted from that of a single XPD helicase in the absence or presence of many RPA2 molecules in solution (see Materials and methods).

*Figure 1d* shows representative traces of a single XPD protein unwinding in increasing [RPA2] at a force of 12 pN. As previously reported, XPD unwound in repeated 'bursts', comprising cycles of forward unwinding motion followed by backward rezipping (*Qi et al., 2013*), repeating a number of times (5 ± 1) until the protein dissociated (e.g., *t* = 95 s for 0 nM RPA2 or *t* = 160 s for 2 nM RPA2 in *Figure 1d*). Unwinding never resumed once a dissociation event occurred, indicating that the single molecule of XPD was flushed away in the flow chamber. In the absence of RPA2, we observe that the processivity—the maximum hairpin position reached by XPD during any one burst—was low, on average 15–20 out of 89 bp, consistent with our prior work (*Qi et al., 2013*). With increasing [RPA2], XPD unwound farther into the hairpin, occasionally unwinding the entire 89 bp hairpin stem (e.g., *t* = 20 s for 5 nM RPA2 or *t* = 30 s, 70 s for 50 nM RPA2 in *Figure 1d*).

Each burst represents an attempt by one XPD molecule to unwind the hairpin, and we found that processivity varied greatly from burst to burst. Each burst consisted of varying extents of duplex unwinding followed by rezipping, often to the base of the hairpin. As shown in the representative trace in *Figure 2a*, several molecular events comprised backward motion: XPD backstepping or backsliding via temporary disengagement from the unwinding DNA strand (*Figure 2b*, diagrams 1a and 1b), or translocation on the other hairpin strand away from the fork junction (2 and 3), all leading to DNA rezipping under its regression force. An example of the latter occurred when XPD completely unwound the hairpin stem and translocated past the dT tetraloop cap onto the opposing strand, allowing the stem to rezip gradually in the protein's wake (*Figure 2b*, diagram 2). In a contrasting example, we observed gradual rezipping of the duplex mid-hairpin, which we attributed to XPD disengaging from its DNA strand to switch to the opposing strand (*Figure 2b*, diagram 3), a behavior reported for other helicases (*Dessinges et al., 2004*; *Comstock et al., 2015*). This interpretation is corroborated by the observation that gradual rezipping was typically followed by a stall at ~10 bp (*Figure 2a*, *t* = 58 s) the size of XPD's footprint, consistent with the protein stalling at the abasic site on the 3′ strand at the base of the hairpin. Subsequent unwinding bursts were presumably due to XPD strand-switching back to the original strand. In contrast, rapid rezipping events due to backsliding usually occurred to the base of the hairpin (*Figure 2a*, *t* = 44 s), consistent with the protein remaining on the hairpin 5′ strand.

We analyzed the processivity of each burst at 12 pN as a function of [RPA2], as shown in the scatter plot in *Figure 2c*. In the absence of RPA2, nearly all the individual burst processivities cluster below 25 bp, with a small fraction (<10%) extending further. When unwinding in the presence of RPA2, the average burst processivity increases with [RPA2] from 19 bp at 0 nM RPA2 to 35 bp at 50 nM RPA2. However, the distributions of burst processivity at each RPA2 concentration show a significant fraction at relatively low processivity, $\leq 25\,bp$, persisting across all concentrations. As [RPA2] increases, an increasing fraction of bursts fall into a higher processivity tail of the distribution, which includes many cases of complete unwinding of the 89 bp hairpin. The distribution of burst processivity for all [RPA2] (*Figure 2d*) shows a large population of bursts with average processivity of 15–20 bp, followed by a long tail with processivity extending from 25 to 89 bp. (See *Table 1* for data trace statistics.)

We also analyzed the effect of force on XPD and its enhancement by RPA2. *Figure 2—figure supplement 1* shows a scatter plot of burst processivities at 9 pN for XPD by itself and in the presence of 10 nM RPA2, a concentration at which most of the processivity enhancement is already detected at 12 pN (*Figure 2c*). A similar pattern of enhancement is obsrved at 9 pN. In the absence of RPA2, XPD burst processivities all cluster below 25 bp, whereas in the presence of RPA2 a significant fraction of bursts (20%) fall in the high-processivity tail of the distribution (25–89 bp). Although the overall processivity is lower at 9 pN compared to 12 pN, consistent with a previous single-molecule study (*Qi et al., 2013*), the average is enhanced by the same ~1.9 factor by RPA2.

## Melting of the hairpin by RPA2 does not aid XPD unwinding

We next asked by what mechanism RPA2 increases the processivity of XPD. One possibility is that RPA2 helps destabilize the duplex, assisting XPD in unwinding. In such a duplex melting mechanism

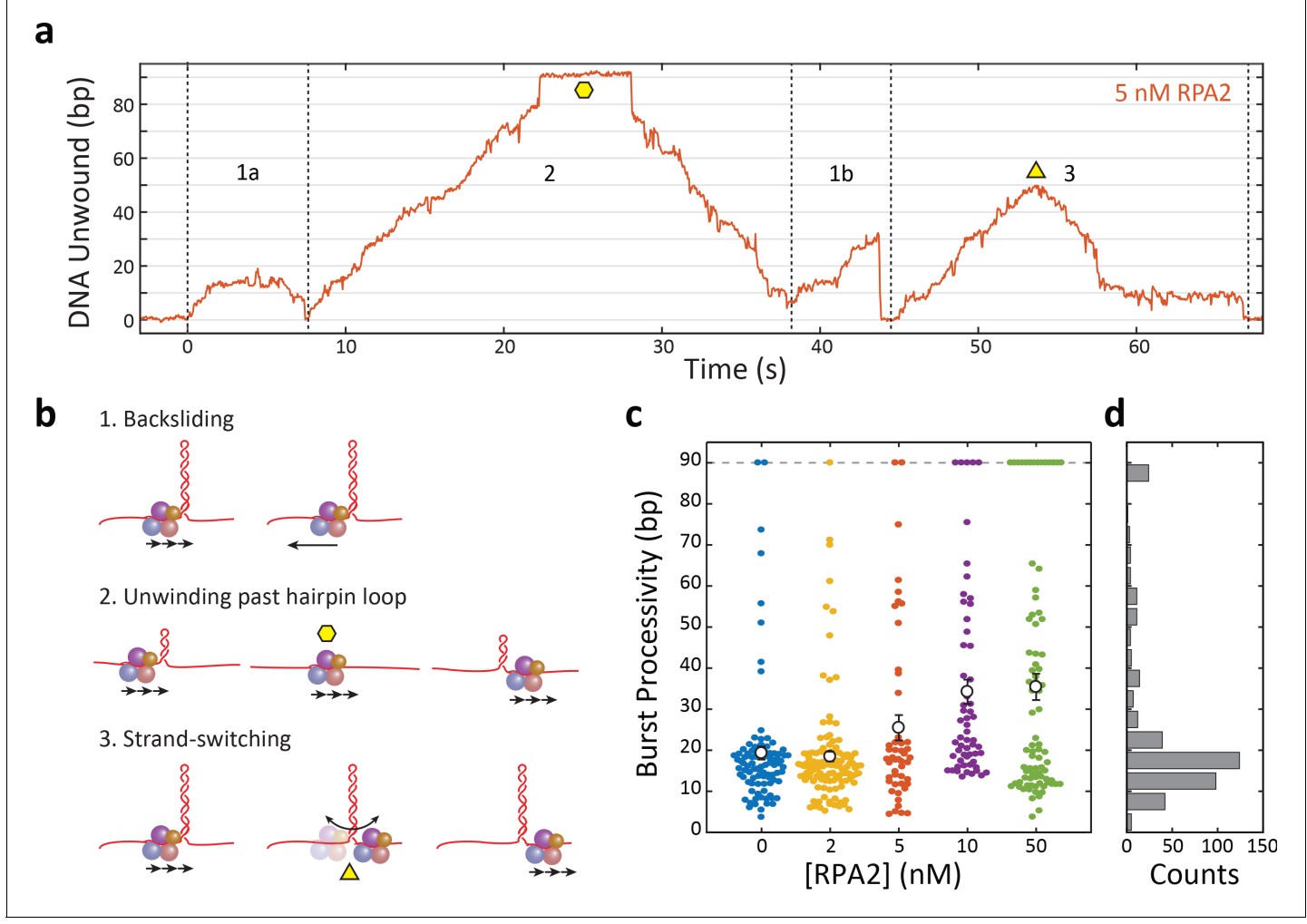

**Figure 2.** Xeroderma pigmentosum group D (XPD) unwinds in bursts of varying processivity whose average increases with replication protein A 2 (RPA2) concentration. (**a**) Representative trace of a single molecule of XPD unwinding in the presence of RPA2 (5 nM) at constant force (*F* = 12 pN). One XPD exhibits repetitive bursts of activity, making multiple attempts to unwind hairpin DNA. Processivity can vary widely from burst to burst. Example time trace with (1a) 20 bp- and (1b) 30 bp-processivity bursts composed of forward unwinding followed by backsliding to the hairpin base; (2) a high-processivity burst during which XPD completely unwinds the 90 bp hairpin past the hairpin loop (time point indicated by yellow hexagon) and translocates on the opposing strand, allowing the hairpin to rezip; and (3) a 50 bp-processivity burst during which XPD unwinds and switches strand mid-hairpin (indicated by yellow triangle), allowing the hairpin to rezip. (**b**) Schematics representing the behaviors in (**a**). (**c**) Processivity of each burst (colored circles) vs. RPA2 concentration. The mean processivity (open circles) increases with RPA2 concentration. Error bars represent s.e.m. (**d**) Histogram of all burst processivities at all RPA2 concentrations.

The online version of this article includes the following source data and figure supplement(s) for figure 2:

**Source data 1.** XPD burst processivity vs. [RPA2].

**Figure supplement 1.** Xeroderma pigmentosum group D (XPD) burst processivity increases with replication protein A 2 (RPA2) concentration at a force of 9 pN.

(*Pugh et al., 2008*), RPA2 would presumably bind at the ssDNA–dsDNA junction and melt several base pairs ahead of XPD. Because XPD relies heavily on thermal fluctuations to help it break the base pairing bonds ahead of it (*Qi et al., 2013*), RPA2 lowering the energy barrier to unzipping the duplex could plausibly enhance XPD's unwinding activity.

As shown in *Figure 3a*, RPA2 is capable of transiently destabilizing hairpin DNA under force of 12 pN. In experiments with RPA2 but without XPD, short melting events are observed, corresponding to the hairpin opening by ~5–6 bp, comparable to the footprint of RPA2 (*Pugh et al., 2008*), and reannealing shortly thereafter (~20 ms). (Such transient melting is also observed in *Figure 1d* after XPD dissociation [e.g., at *t* > 110 s for 50 nM RPA2].) These melting events become more frequent

**Table 1.** Data statistics.

| Force (pN) | 7–8 | | 9 | | | | 12 | | | | | | |
|---|---|---|---|---|---|---|---|---|---|---|---|---|---|
| XPD | wt | | wt | | H202A | | wt | | | | | H202A | |
| [RPA2] (nM) | | | 0 | 10 | 0 | 10 | 0 | 2 | 5 | 10 | 50 | 0 | 10 |
| [gp32] (nM) | 0 | 250 | | | | | | | | | | | |
| No. of XPD molecules | 10 | 8 | 6 | 25 | 11 | 11 | 20 | 19 | 27 | 20 | 21 | 29 | 16 |
| No. of bursts | 141 | 23 | 54 | 74 | 38 | 67 | 94 | 123 | 79 | 60 | 82 | 123 | 79 |
| No. of bursts/XPD | 14.1 | 2.9 | 9 | 3 | 3.5 | 6.1 | 4.7 | 6.5 | 2.9 | 3.0 | 3.9 | 4.2 | 4.9 |
| No. of low-processivity bursts | 141 | 23 | 54 | 60 | 36 | 63 | 86 | 109 | 61 | 31 | 45 | 46 | 36 |
| No. of high-processivity bursts | 0 | 0 | 0 | 14 | 2 | 4 | 8 | 14 | 18 | 29 | 37 | 77 | 43 |

XPD: xeroderma pigmentosum group D; RPA2: replication protein A 2; gp32: gene protein 32.

as the concentration of RPA2 is increased (*Figure 3—figure supplement 1*), in support of melting being RPA2-mediated. Analyzing the times between melting events, we estimate an effective second-order rate constant of $(1.2 \pm 0.3) \times 10^8 \text{ M}^{-1} \text{ s}^{-1}$ for binding.

RPA2-mediated melting events are also observable as XPD unwinds in the presence of RPA2 (*Figure 3b*). Such events are identifiable over XPD unwinding dynamics due to their characteristic size (~5–6 bp) and short lifetime (~20 ms) (*Figure 3b*, inset; see Materials and methods and *Figure 3—figure supplement 2*). While some false positives are detected in the absence of RPA2, they are uncommon and the frequency of detected melting events increases with [RPA2] (*Figure 3c*), confirming their connection to RPA2 activity. This finding indicates that RPA2 is capable of binding at the same fork junction as XPD and melting DNA ahead of the helicase. However, the question remains whether RPA2 transient melting aids XPD in unwinding.

To answer this question, we determined how RPA2-mediated melting events impact the forward progress of the helicase. If duplex destabilization aids in unwinding, we suspected XPD may be more likely to advance along the DNA during melting events. For example, RPA2 melting could maintain the DNA fork open by ~5 bp, allowing XPD to translocate rapidly to catch up to the fork position and then to take the lead unwinding at the fork junction. However, analyzing the occurrences of ~5 bp melting events followed by XPD unwinding by its step size of 1 bp (*Qi et al., 2013*; *Figure 3—figure supplement 2*; light blue box), we found that such pairs of events become less frequent as RPA2 concentration increases ($8.0 \pm 0.5\%$ in the absence of RPA2 vs. $4.0 \pm 0.2\%$ at 50 nM RPA2), inconsistent with this mechanism.

To determine if XPD could advance during RPA2-mediated melting, we examined the position of the helicase on the hairpin prior to each RPA2 melting event and after the duplex reannealed. We identified many ($N \approx 400$) individual melting events during XPD unwinding and aligned them along the position axis such that each event began at 0 bp before melting occurred (*Figure 3d*, schematic). We then independently aligned the melting and reannealing transition of each event in time at $t = t_{melt}$ and $t = t_{anneal}$, respectively (*Figure 3d*, 'before' and 'after' schematic). As shown in *Figure 3e*, the resulting probability density of these aligned traces reveals the average position of XPD prior to and following an RPA2 melting event (see Materials and methods). After a melting event, we find that the hairpin is most likely to reanneal all the way back to the same position as before the event, indicating that XPD has not moved from its initial position. Importantly, this result shows that there is no significant forward movement of XPD as a result of RPA2 duplex melting, and that XPD does not exploit RPA2's melting activity to enhance its own unwinding. *Figure 3f* shows a control in which the alignment process above was repeated at random points during unwinding, each point paired with one $t = 0.06$ s after, corresponding to the time window over which we searched and identified RPA2 melting events (see Materials and methods). Again, XPD on average shows no net progress in this amount of time. Comparing *Figure 3e and f* shows that the RPA2-mediated melting events have little to no impact on XPD movement.

Our data collected at lower forces further disfavor a melting mechanism. As force is decreased, RPA2-mediated melting occurs less frequently, with almost no melting detected at 9 pN and below (*Figure 3—figure supplement 3*). If RPA2 melting were necessary to increase the unwinding

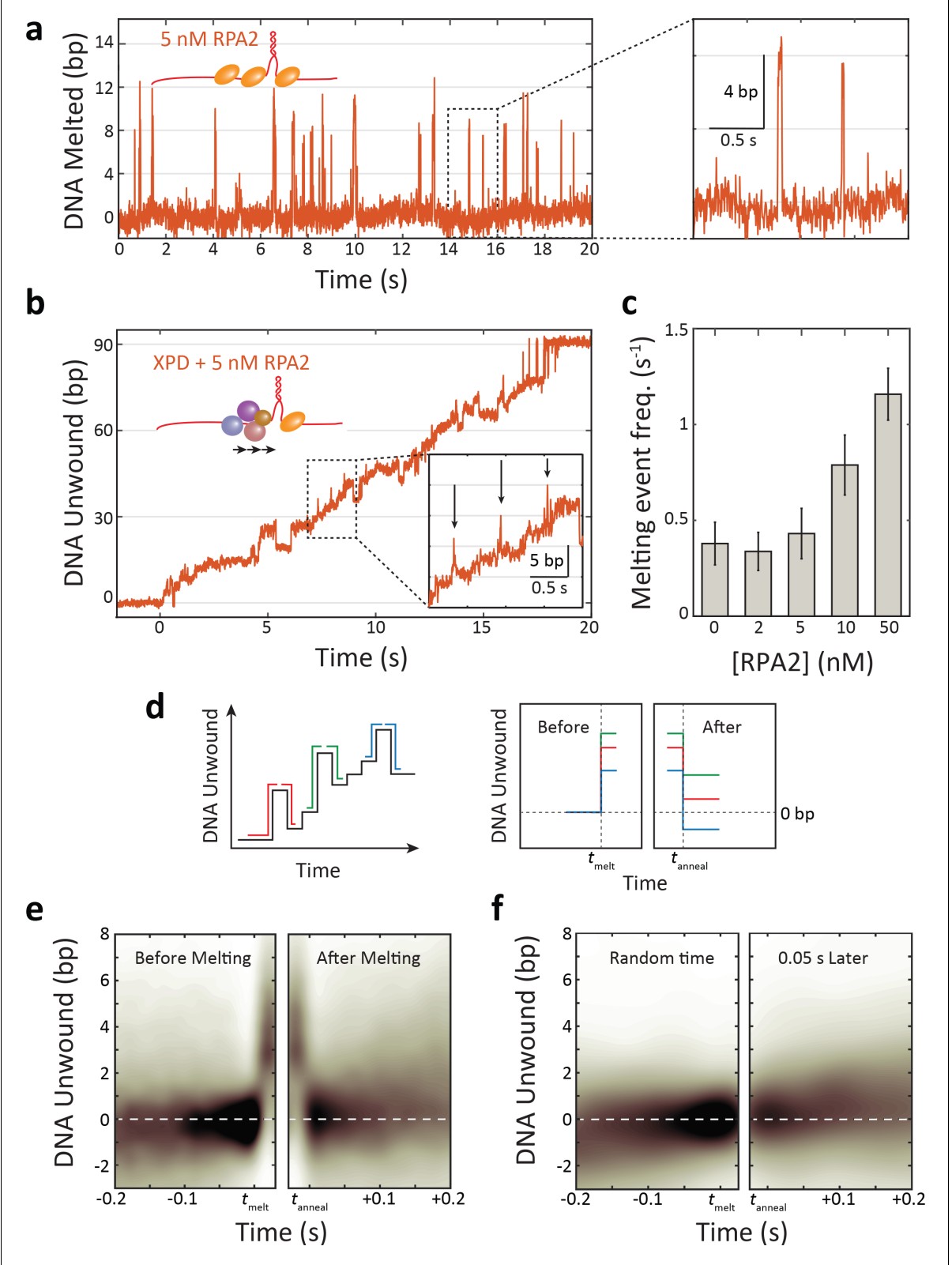

**Figure 3.** Replication protein A 2 (RPA2) transiently melts hairpin duplex but does not assist xeroderma pigmentosum group D (XPD) unwinding. (a) Representative time trace of RPA2 transiently destabilizing hairpin dsDNA at a constant force ($F$ = 12 pN). Inset: RPA2 melts ~8 bp, which then rapidly reanneals (see **Figure 3—figure supplement 1**). (b) RPA2 is able to melt dsDNA at a fork occupied with XPD. Inset: transient RPA2-like melting events during XPD unwinding (see **Figure 3—figure supplement 2**). (c) The frequency of RPA2 melting events increases with RPA2 concentration. Error bars

*Figure 3 continued on next page*

Figure 3 continued

represent s.e.m. (d) Analysis of RPA2 melting events and their effect on XPD unwinding. The schematic shows all RPA2 melting events identified and divided into 'melting' and 'reannealing' transitions. Each transition is aligned temporally such that all melting transitions begin at the same time $t_{melt}$ and corresponding reannealing transitions end at the same time $t_{anneal}$. Both types of transitions are aligned spatially relative to the starting extension before melting ($x = 0$). (e) Probability distribution of all aligned RPA2 melting events. Although RPA2 melts an average of 5 bp of hairpin DNA (left), it reanneals by the same amount (right), and there is no net progress of XPD due to melting. (f) Probability distribution at a random time point (left) and 0.06 s later (right). Net progress of XPD after an RPA2 melting event is indistinguishable from net progress at a random time point. Probability distributions were obtained using kernel density estimation (see Materials and methods).

The online version of this article includes the following source data and figure supplement(s) for figure 3:

**Source data 1.** RPA2 melting statistics and XPD burst processivity vs. [gp32].
**Figure supplement 1.** Replication protein A 2 (RPA2) transiently melts DNA under force.
**Figure supplement 2.** Detecting replication protein A 2 (RPA2) melting events during xeroderma pigmentosum group D (XPD) unwinding.
**Figure supplement 3.** Replication protein A 2 (RPA2) transient melting of DNA depends on force.
**Figure supplement 4.** T4 gene protein 32 (gp32) transiently melts hairpin duplex but does not assist xeroderma pigmentosum group D (XPD) unwinding.

processivity of XPD, we would expect little to no change in activity at these forces. However, XPD activity was enhanced by RPA2 at 9 pN (*Figure 2—figure supplement 1*), with a significantly higher fraction of bursts with processivities in the long tail extending from 25 to 89 bp.

Lastly, we measured XPD unwinding in the presence of bacteriophage T4 gene protein 32 (gp32), a non-cognate single-stranded DNA binding protein capable of destabilizing the DNA duplex (*Pugh et al., 2008*; *Pant et al., 2003*). Like RPA2, gp32 is a monomeric protein containing a single OB-fold and thus possesses a similar footprint as RPA2 (*Robbins et al., 2005*; *Shamoo et al., 1995*; *Theobald et al., 2003*). Prior ensemble studies showed that gp32 can melt short forked duplexes, but enhances XPD unwinding to a much lower extent than RPA2 under the same conditions (*Pugh et al., 2008*). If transient duplex melting were sufficient to enhance XPD processivity, we would expect gp32 to yield similar enhancements under conditions where it melts the hairpin to a similar extent as RPA2. We determined that at forces of 7–8 pN and a concentration of 250 nM gp32 melts the hairpin transiently by ~5–10 bp, comparable to RPA2 at 12 pN (*Figure 3—figure supplement 4a*). We estimate that the extent of melting by gp32 under these conditions is equivalent to that of 10–35 nM RPA2 at 12 pN. (Hairpin DNA is transiently melted 2.5% and 7.0% of the time at 10 and 35 nM RPA2, respectively, compared to 4.1% of the time for gp32.) However, gp32 has a minimal impact on XPD processivity under these conditions. We observed no bursts in the >25 bp, high-processivity tail of the distribution (*Figure 3—figure supplement 4b*), in marked contrast to the corresponding conditions with RPA2 (*Figure 2d*). Taken together, these data strongly indicate that enhancement of XPD processivity by RPA2 is mediated by a mechanism other than duplex melting.

## XPD exhibits two distinct types of activity, the fraction of which depends on RPA2

The burst processivities at different [RPA2] (*Figure 2c*) show that a persistent and significant fraction of bursts exhibit relatively low processivity, $\lesssim 25\,bp$. Bursts with higher processivity extending from 25 to 89 bp also exist at all [RPA2], with their number apparently increasing as [RPA2] increases. An important question is whether there exist any characteristic differences between bursts during which more or less DNA is unwound.

To answer this question, we grouped bursts into two categories: 'high processivity' or 'low processivity' depending on whether or not a threshold of 25 bp was exceeded, this value chosen based on where the tail in the distribution of processivities occurred (*Figure 2c*). *Figure 4a* shows the unwinding portions of many bursts at a force of 12 pN aligned to begin at the same time ($t = 0$), color coded by [RPA2] and processivity category (light color for low processivity; dark color for high processivity; see *Table 1* for the number of data traces). When grouped in this manner, some notable features emerge. The fraction of bursts categorized as high processivity increases clearly with RPA2 concentration (*Figure 4a*) saturating to ~50% at RPA2 concentrations >10 nM (*Figure 4b*). Furthermore, long stalls (sometimes exceeding tens of seconds in duration) around 10–15 bp are evident at all concentrations and in both high- and low-processivity traces. This stalling

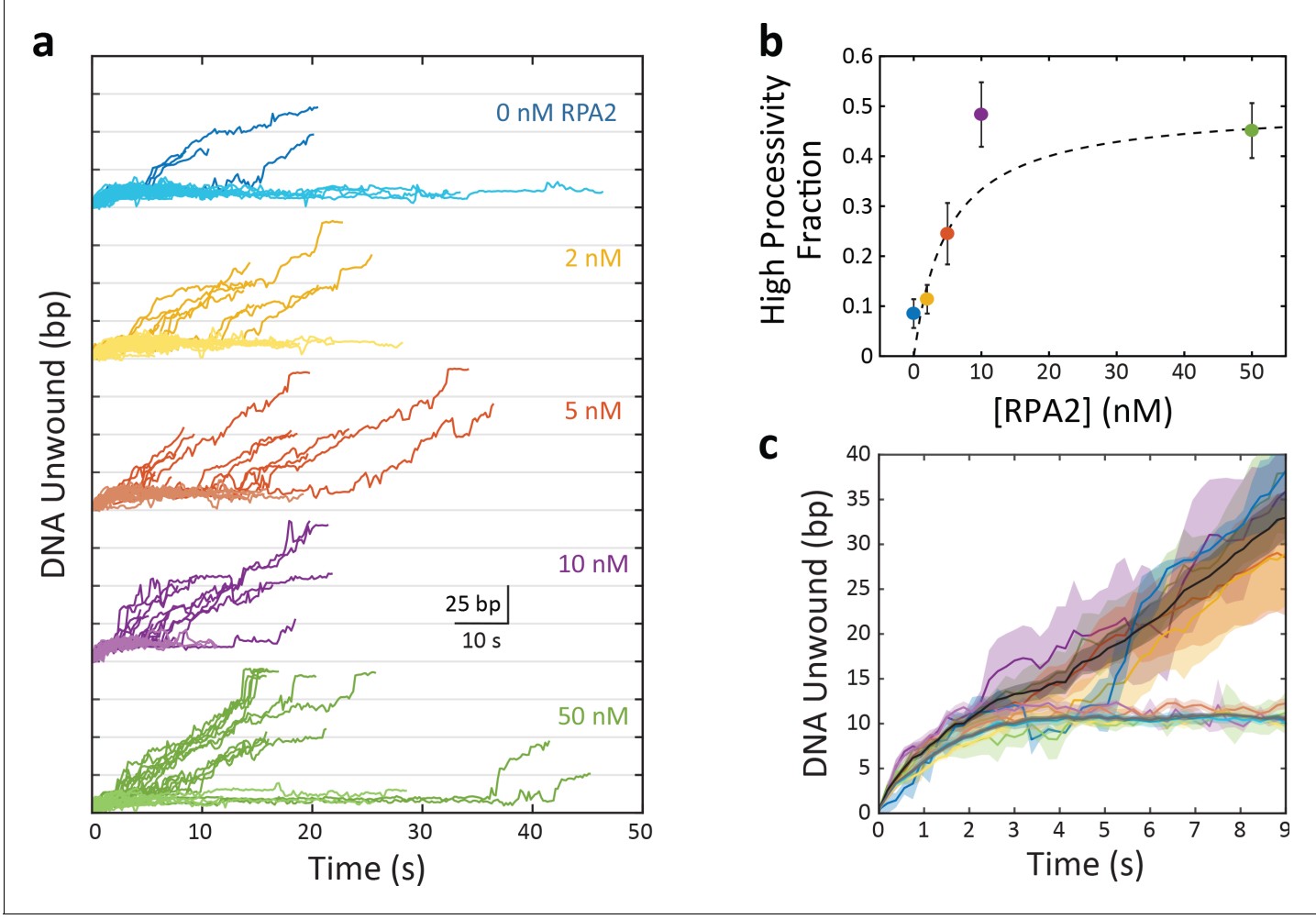

**Figure 4.** Xeroderma pigmentosum group D (XPD) exhibits two burst types, the fraction of which is replication protein A 2 (RPA2) dependent. (**a**) Plot of XPD unwinding bursts, aligned to start at $t = 0$ and grouped by RPA2 concentration (colored traces). XPD unwinding bursts come in two types: low processivity, never unwinding more than 25 bp (light colors); and high processivity, unwinding more than 25 bp (dark colors). (**b**) The fraction of high-processivity bursts (>25 bp) increases with RPA2 concentration. Fit to model described in the text (dashed line; see Materials and methods). (**c**) Averages of all low-processivity bursts (light colored lines) and all high-processivity bursts (dark colored lines) at each RPA2 concentration. Comparison to averages of low- and high-processivity burst types over all RPA2 concentrations (dark gray and black lines, respectively). Shaded regions represent s.e.m. throughout. Unwinding behavior within each burst category remains the same over all RPA2 concentrations.

The online version of this article includes the following source data and figure supplement(s) for figure 4:

Source data 1. High-processivity fraction and burst duration vs. [RPA2], and XPD speed vs. position.

Figure supplement 1. Unwinding velocity varies with processivity type but not replication protein A 2 (RPA2) concentration.

Figure supplement 2. Xeroderma pigmentosum group D (XPD) backward motion varies with processivity type but not replication protein A 2 (RPA2) concentration.

behavior has been reported previously (*Qi et al., 2013*) and attributed to XPD's encounter with GC-rich regions—which are more difficult to unwind due to higher base-pairing energy—at this position in the hairpin. All low-processivity bursts exhibit these stalls, after which XPD either backslides to the base of the hairpin or dissociates, ending the burst. However, we have observed XPD to continue unwinding past 25 bp after a long stall and be categorized as 'high processivity'.

Importantly, we do not observe significant differences between bursts of the same processivity category obtained at different RPA2 concentrations. *Figure 4c* shows all unwinding bursts at 12 pN for high- and low-processivity types averaged together at each RPA2 concentration (color coded the same as in *Figure 4a*) and across all RPA2 concentrations (black and gray lines, respectively). The averaged unwinding traces exhibit the same behavior independent of RPA2 concentration. Low-

processivity bursts all display stalls at ~10 bp, whereas high-processivity bursts unwind past this region. Burst-to-burst differences are reflected by the shaded areas and are consistently smaller than the differences between the two processivity categories. This finding is corroborated when analyzing the velocity of XPD. *Figure 4—figure supplement 1* shows the average unwinding velocity determined separately for low- and high-processivity bursts at each RPA2 concentration as a function of XPD's position on the hairpin. While XPD on average slows down near 10 bp due to the high GC content of this section of the hairpin, the velocity of high-processivity bursts is consistently higher than for low-processivity bursts at this position. On the other hand, differences in velocity are insignificant across varying [RPA2] within the same processivity category.

We also find that both low- and high-processivity burst durations remain constant with RPA2 concentration (*Figure 4—figure supplement 2*, light-colored and open data points, respectively). The more processive bursts tend to have longer durations since XPD travels a longer distance on DNA, and processivity increases with [RPA2]. The increase in burst duration observed when combining both types of bursts is explained simply from the increase in the fraction of high-processivity bursts with increasing [RPA2].

Our results show that high-processivity unwinding in the absence of RPA2 is indistinguishable from that in the presence of RPA2 and likewise for low-processivity unwinding. All parameters we quantified by burst in the same processivity category are independent of [RPA2]. Instead, [RPA2] increases the probability of high-processivity bursts. These findings suggest that high- and low-processivity unwinding correspond to intrinsic states of XPD, with RPA2 increasing the likelihood of XPD being in a state competent for high-processivity unwinding.

## A mutant of XPD recapitulates the behavior of wild-type XPD in the presence or RPA2

In a previous study, point mutations in the homologous *Thermoplasma acidophilum* XPD were identified that yielded increased unwinding activity and force generation (*Pugh et al., 2012*). We enquired if mutants of *Fac*XPD could act similarly to the high-processivity state proposed above in overcoming energetic barriers in the DNA sequence (such as the high-GC section of the hairpin near 10 bp) and what the effect of RPA2 on their activity would be. We thus synthesized the mutant *Fac*XPDH202A, which targeted the site analogous to H198 in *Tac*XPD that had been shown to elicit the largest positive effect on unwinding activity (*Pugh et al., 2012*). This residue is located on domain HD1, within a secondary binding site thought to interact with the translocating strand of DNA in the groove between HD1 and the FeS domains (see *Figure 1a*, black DNA contacts behind the FeS domain). It is distinct from the residues shown to be responsible for XPD damage verification (*Mathieu et al., 2013*) (see Discussion).

*Figure 5a* shows representative time traces of XPDH202A unwinding alone (cyan) and in the presence of 10 nM RPA2 (magenta) at 12 pN of force. Like wild-type (wt) XPD, XPDH202A also unwinds DNA in repeated bursts. However, on its own, XPDH202A exhibits higher processivity and can regularly unwind the entire hairpin during its multiple bursts on the DNA hairpin. As shown in the scatter plot in *Figure 5b*, the average burst processivity of XPDH202A greatly exceeds that of wt XPD and is also slightly higher than that of wt XPD in the presence of 10 nM RPA2. Similarly, the fraction of high-processivity bursts for XPDH202A is much higher than for wt XPD and comparable to that for wt XPD with RPA2 (*Figure 5c*). Interestingly and unlike wt XPD, the unwinding processivity of XPDH202A does not increase appreciably in the presence of 10 nM RPA2 (*Figure 5b, c*). We observed similar trends at a lower force of 9 pN (*Figure 5—figure supplement 1*). Overall, the processivity of XPDH202A (alone or in the presence of RPA2) is similar to that of wt XPD with RPA2, suggesting that RPA2 elicits the same effect on XPD as the H202A mutation.

## Discussion

Our single-molecule measurements of XPD with RPA2 provide important clues to decipher the underlying mechanism of processivity enhancement. We have already explored some mechanisms in which RPA2 binding to DNA could enhance XPD activity. Our results in *Figure 3* (and also *Figure 2—figure supplement 1* and *Figure 3—figure supplement 4*) rule out RPA2 melting of DNA as a plausible mechanism. The data also disfavor a sequestration mechanism, in which RPA2 binding to ssDNA behind XPD inhibits retrograde motion. By acting as a physical barrier against XPD

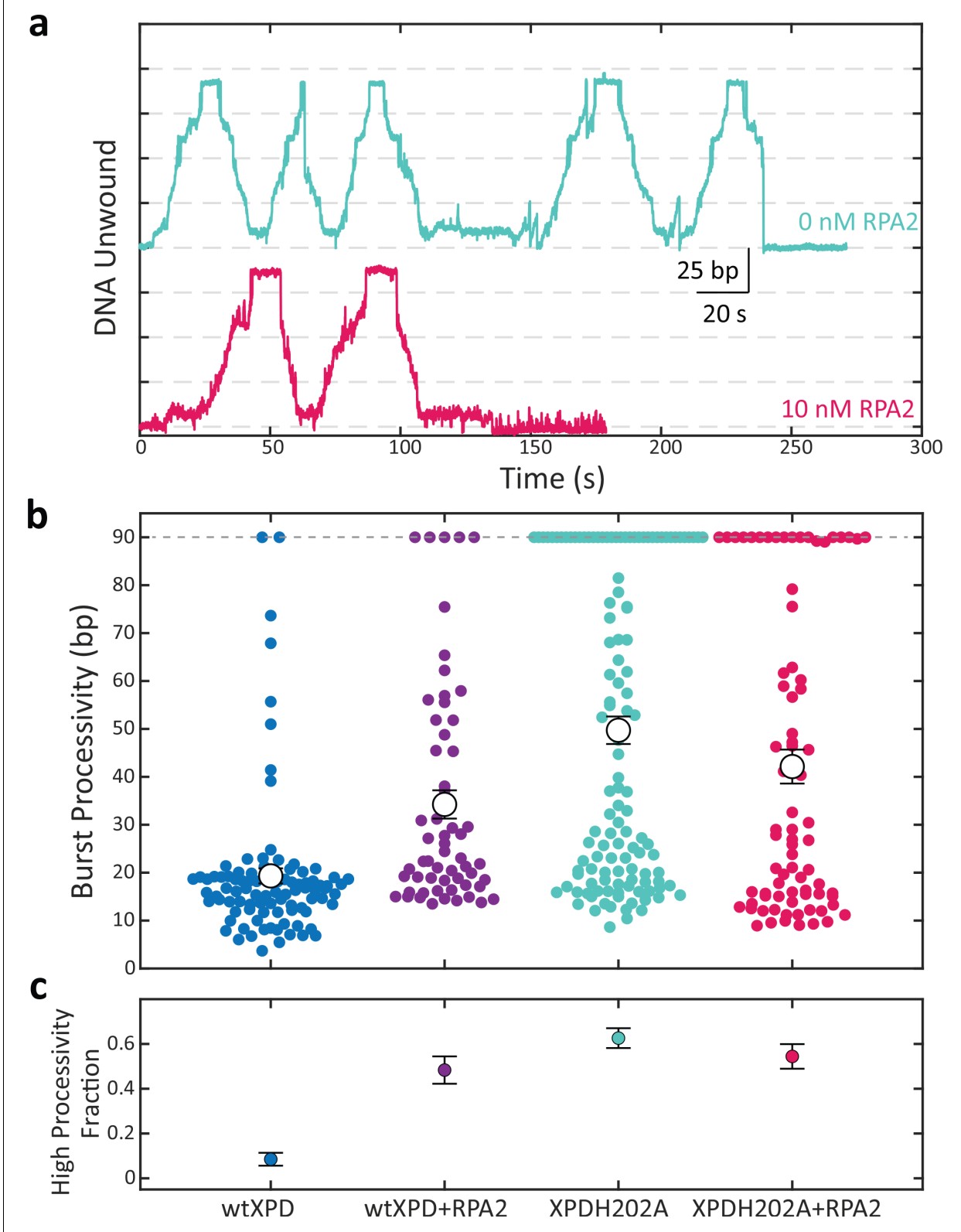

**Figure 5.** A point mutation in xeroderma pigmentosum group D (XPD) enhances its processivity similarly to replication protein A 2 (RPA2). (a) Representative traces of a single molecule of the mutant XPDH202A unwinding at constant force ($F$ = 12 pN) alone (cyan) and in the presence of RPA2 (10 nM; magenta). (b) Processivity of individual bursts (colored circles) at a constant force of 12 pN for wild-type XPD alone (blue) and with 10 nM RPA2 (purple), and for XPDH202A alone (cyan) and with 10 nM RPA2 (magenta). The H202A mutation increases XPD's mean processivity (open circles);

*Figure 5 continued on next page*

*Figure 5 continued*

addition of RPA2 does not enhance processivity further. (**c**) The fraction of high-processivity bursts (>25 bp) corresponding to (**b**). The H202A mutation increases the fraction of high-processivity bursts to a level similar to that of XPD with RPA2. Error bars represent s.e.m. throughout.

The online version of this article includes the following source data and figure supplement(s) for figure 5:

**Source data 1.** XPD and mutant burst processivity vs. [RPA2], and complex formation statistics .
**Figure supplement 1.** A point mutation in xeroderma pigmentosum group D (XPD) enhances its processivity similarly to replication protein A 2 (RPA2) at a force of 9 pN.
**Figure supplement 2.** Test of stable complex formation.

backstepping or backsliding (*Figure 4—figure supplement 2a*), two behaviors often exhibited by XPD during unwinding (*Qi et al., 2013*), ssDNA sequestration could in principle enhance processivity. However, identifying XPD backsteps and backslides in unwinding traces (e.g., *Figure 3—figure supplement 2*) shows that RPA2 did not reduce their frequency. There is a marginal increase in the fraction of backsteps (which we defined as steps between −2 and 0 bp) from 19 ± 1% to 25 ± 1% and almost no change in the fraction of backslides (defined as steps smaller than −2 bp) from 18 ± 1% to 19 ± 1% as RPA2 increased from 0 to 50 nM. Furthermore, since each burst involves XPD-mediated DNA unwinding followed by rezipping to the base of the hairpin, the duration of a burst provides information on the amount of retrograde motion. We expect bursts with less frequent backsteps and backslides to be longer in duration since XPD would unwind further into the hairpin and thus take longer to return to the hairpin base. However, as shown in *Figure 4—figure supplement 2b*, we do not observe the mean duration of either the low- or high-processivity bursts to increase with RPA2, as predicted by a sequestration mechanism.

Alternately, RPA2 may enhance unwinding through direct interactions with XPD (*Figure 5—figure supplement 2*). Direct RPA–helicase interactions have been detected for the superfamily 2 human helicases WRN, BLM, RECQ, FANCJ, and SMARCAL1 and shown to increase helicase activity (*Cui et al., 2004*; *Gupta et al., 2007*; *Bétous et al., 2013*; *Brosh et al., 1999*; *Brosh et al., 2000*; *Doherty et al., 2005*). The inability of the non-cognate T4 gp32 to enhance XPD unwinding (*Pugh et al., 2008* and *Figure 3—figure supplement 4*) strongly suggests that specific interactions between XPD and RPA2 are essential. While no strong interactions between *Fac*XPD and *Fac*RPA2 have been reported in solution (*Pugh et al., 2008*), formation of an XPD–RPA2 complex may require pre-assembly on DNA.

Our measurements of the mutant XPDH202A point to the potential nature of specific XPD–RPA2–DNA interactions. As shown in *Figure 6a*, XPD has several secondary DNA binding sites, and the residue H202 lies within one site on the HD1 domain, in the groove between the HD1 and FeS domains (*Kuper et al., 2012*; *Pugh et al., 2012*). XPD–ssDNA contacts at this site are believed to be important for DNA fork positioning and to play an inhibitory role, acting as a throttle to unwinding (*Pugh et al., 2012*). Disruption of contact points by mutagenesis in both *Tac*XPD (*Pugh et al., 2012*) and *Fac*XPD (*Figure 5*) enhances unwinding. Structural analysis of XPD and homologs maps ~5 nt of the translocating DNA strand to this secondary site between the DNA fork at the FeS domain and the entry pore into the motor core (*Qi et al., 2013*; *Kokic et al., 2019*; *Pugh et al., 2012*; *Figure 6a*). We propose that RPA2 binding of these 5 nt of ssDNA could interfere with the inhibitory XPD–ssDNA contacts at this secondary site and activate XPD unwinding to a similar extent as mutation. As we have shown, RPA2 can bind at a DNA junction occupied with XPD (*Figure 3*), and the 5 nt matches the footprint of RPA2 (*Pugh et al., 2008*). Such a mechanism is consistent with our observation that XPDH202A processivity is comparable to that of wt XPD in the presence of RPA2, and that RPA2 does not enhance XPDH202A processivity further (*Figure 5*, *Figure 5—figure supplement 1*). In further support of this model, prior single-molecule measurements (*Qi et al., 2013*) showed that individual XPD exhibits 5 bp backward/forward step pairs (see also *Figure 3—figure supplement 2*; dark blue box), attributed to the transient release/recapture of 5 nt of ssDNA from the secondary site. In the presence of RPA2, we observe the frequency of such events to decrease (*Figure 3—figure supplement 2*; compare dark blue box for 0 nM to 50 nm RPA2), consistent with the idea that RPA2 may affect DNA contacts to this regulatory site. Lastly, the fact that gp32 has no effect on XPD despite having a similar footprint as RPA2 indicates that binding of

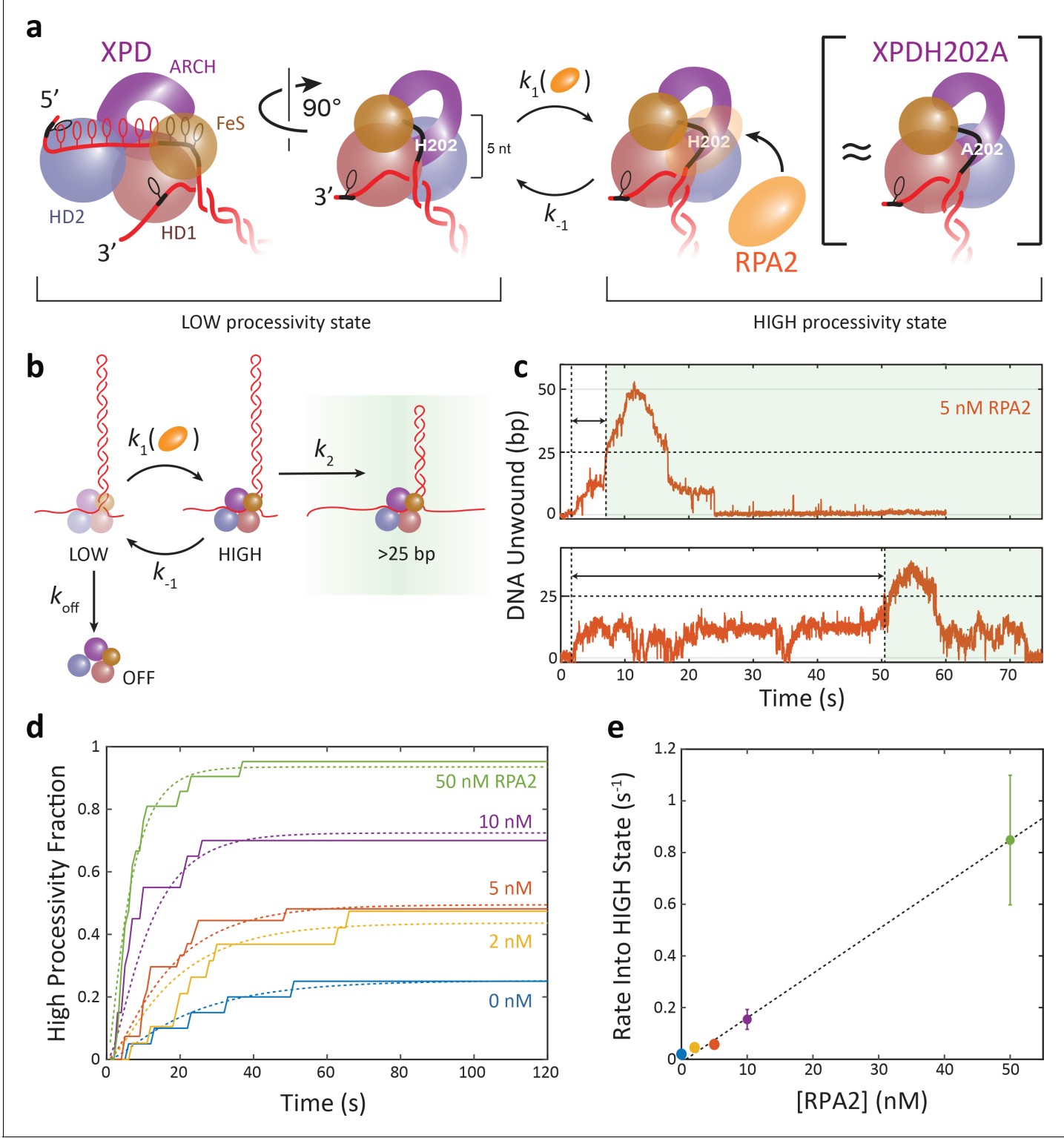

**Figure 6.** Replication protein A 2 (RPA2) activates a high-processivity state of xeroderma pigmentosum group D (XPD). (**a**) Model of XPD enhancement by RPA2. Schematic of XPD–DNA complex (left; side and front views), with 5 nt of ssDNA (black) bound at a regulatory secondary binding site on HD1 that contains H202. XPD can adopt one of two intrinsic states—a low (left) and high (right) processivity state—that correspond to different DNA fork conformations. Interconversion between both states occurs either spontaneously or through RPA2 binding of the 5 nt at the regulatory site. XPDH202A adopts a similar conformation (right). (**b**) Kinetic model of XPD processivity and the effect of RPA2. XPD interconverts between low and high states with rates $k_1$ and $k_{-1}$, and RPA2 shifts the equilibrium toward the high state. XPD can dissociate from DNA only from the low state with rate $k_{off}$. Only in the

*Figure 6 continued on next page*

Figure 6 continued

high state is XPD able to unwind in excess of 25 bp, which occurs at a rate $k_2$. Once an XPD molecule unwinds >25 bp, it is scored as being in the high-processivity state. (c) Representative traces of XPD exhibiting high-processivity unwinding. The time $t_{>25}$ denotes the first time at which XPD crosses the 25 bp threshold. (d) Fraction of all XPD molecules that have reached high processivity (>25 bp) after time $t$ for each RPA2 concentration. The curves are globally fit to the model in (a) using as parameters the rate constants $k_1$, $k_{-1}$, $k_2$, and $k_{off}$. (e) The rate of entry into the high-processivity state, $k_1$, depends linearly on RPA2 concentration. For more details on the kinetic model, see Materials and methods.

The online version of this article includes the following source data for figure 6:

**Source data 1.** High-processivity fraction vs. time and high-processivity state entry rate vs. [RPA2].

ssDNA at the secondary site is insufficient by itself; specific protein–protein contacts between RPA2 and XPD must also be required.

As the above arguments point to direct interactions between XPD and RPA2 mediated by DNA, we tested whether we could pre-form a stable XPD–DNA complex on DNA. We carried out an experiment in which we allowed XPD and RPA2 to bind to DNA in the absence of ATP, then provided ATP for unwinding. Here, we utilized a different flow chamber configuration in which one stream contained XPD and RPA2 (*Figure 5—figure supplement 2b*; 'loading' stream) but no ATP, and another stream contained ATP but no protein ('unwinding' stream). We first placed the DNA hairpin in the protein 'loading' stream for 40–60 s, then moved it into the ATP-containing 'unwinding' stream to observe unwinding. If XPD and RPA2 were to form a complex with higher processivity than XPD alone, we would expect more processive activity at higher RPA2 concentrations as the probability of a complex forming should increase. However, we observed no effect on processivity across RPA2 concentration (*Figure 5—figure supplement 2c*); the fraction of bursts that exhibited high processivity (>25 bp) remained constant. This result indicates that any complex formed would be short-lived and would not survive the move between the protein and ATP streams, which takes up to ~10 s.

Integrating all our results, *Figure 6a* shows our proposed mechanism. XPD can adopt two intrinsic states, one competent for high-processivity unwinding and the other not. We propose that these two states correspond to different conformations of the DNA fork in its interactions with the secondary binding site on XPD. Both states are populated independently of RPA2; highly processive unwinding activity can occur in the absence of RPA2 and is indistinguishable from that detected in the presence of RPA2. RPA2 does not assist XPD unwinding by directly destabilizing the duplex, but rather by transiently shifting the equilibrium toward the state competent for high processivity. RPA2 binding to ssDNA at the fork interferes with its interactions with the secondary site in a similar way as the H202A mutation, activating high-processivity unwinding. In the data we have collected, the two states are effectively hidden, and we can infer their presence only from the processivity exhibited by XPD, that is, whether XPD unwinds beyond a threshold we have selected as 25 bp, past the first high-GC energetic barrier in the hairpin sequence that XPD must overcome. As a result, the correspondence between conformational state and processivity need not be exact.

*Figure 6b* shows a simple kinetic scheme that describes this model quantitatively. XPD can switch between 'low'- and 'high'-processivity states with rates $k_1$ and $k_{-1}$. Here, we assume $k_1$ to be dependent on RPA2 concentration, whereas all other rate constants are independent of RPA2. In the high-processivity state, XPD reaches the 25 bp threshold—and is scored as high processivity—at a rate $k_2$, corresponding to the time XPD takes to unwind 25 bp, which encompasses all forward and backward steps to this position. Finally, we assume that XPD can dissociate from the DNA in the low-processivity state with rate constant $k_{off}$. Given this kinetic scheme, we derive an expression for the probability, $P_{>25}(t)$, that a given XPD molecule unwinds >25 bp by time $t$ (see Materials and methods), which can be compared directly to experimental data (*Figure 6d*), obtained from the fraction of XPD molecules that have crossed the 25 bp threshold at each time point (*Figure 6c*).

*Figure 6d* shows a global fit to our data using different values of $k_1$ for each RPA2 concentration but a common set of values of $k_{-1}$, $k_2$, and $k_{off}$. (We considered an alternative model in which XPD can dissociate in both its low- and high-processivity states by introducing an additional rate constant $k'_{off}$; see Materials and methods. The best fit to the data yielded negligible values for $k'_{off}$, indicating that XPD dissociation occurs in its low-processivity state but is unlikely from its high-processivity state.) The parameter values can be found in *Table 2* and are consistent with independent

**Table 2.** Model fit parameters.

| Rate constant | Fit value ($s^{-1}$) |
| --- | --- |
| $k_{-1}$ | $0.18 \pm 0.13$ |
| $k_2$ | $0.170 \pm 0.017$ |
| $k_{off}$ | $0.037 \pm 0.019$ |
| $k'_{off}$ | 0 |
| $k_1$, [RPA2]=0 nM | $0.02 \pm 0.006$ |
| $k_1$, 2 nM | $0.059 \pm 0.017$ |
| $k_1$, 5 nM | $0.065 \pm 0.019$ |
| $k_1$, 10 nM | $0.200 \pm 0.059$ |
| $k_1$, 50 nM | $1.15 \pm 0.39$ |

Error bars represent 95% confidence intervals.
RPA2: replication protein A 2.

measurements. The value obtained for $k_{off}$ predicts that XPD remains bound to DNA for an average lifetime >25 s, consistent with observation. A value of $k_2 = 0.17 \pm 0.02$ $s^{-1}$ matches that expected for XPD to unwind 25 bp at an average speed of 4–5 bp/s (*Figure 4—figure supplement 1a*), a speed consistent with previous measurements (*Qi et al., 2013*). The rate constant $k_{-1}$ in our kinetic model determines the lifetime of the high-processivity state, and its value gives a lifetime of ~8 s, which agrees with interpretations of a short-lived complex lasting <10 s (*Figure 5—figure supplement 2*). The transience of the interaction may also explain why an XPD molecule may revert to low-processivity activity after a high-processivity burst and vice versa. Finally, we find that $k_1$ increases linearly with RPA2 concentration (*Figure 6e*). The linear dependence on [RPA2] suggests that a simple second-order reaction between RPA2 and the XPD–DNA complex shifts the equilibrium to the high-processivity state and rules out mechanisms in which multiple RPA2 bind cooperatively. The second-order rate constant obtained from the slope of $k_1$ vs. [RPA2], $(1.5 \pm 0.5) \times 10^7$ $M^{-1}$ $s^{-1}$, is approximately eight times smaller than our estimated rate constant for RPA2 binding to bare ssDNA (*Figure 3—figure supplement 1*), expected since DNA-bound XPD should partially block RPA2 access to ssDNA. The intercept of $1–2 \times 10^{-2}$ $s^{-1}$ provides a basal rate of interconversion between the two processivity states in the absence of RPA2. This simple kinetic model can also be compared to our other results. The fraction of high-processivity XPD bursts measured vs. RPA2 (*Figure 4c*) is well fit by the model with no additional fitting parameters (see Materials and methods). Likewise, the average burst duration (*Figure 4—figure supplement 2*) is well described by the same model (see Materials and methods).

Helicases are likely to encounter other DNA-bound proteins in the cell, and such molecular associations have the potential to regulate helicase activity. Our results provide new insights into the possible mechanisms by which accessory proteins can affect helicases. We show how a single-stranded DNA binding protein can enhance helicase processivity through transient interactions with the helicase-DNA complex, which can activate a 'processivity switch'. We speculate on the latent conformational states that could be controlled by this switch. The evidence above points to alternate DNA binding configurations (*Qi et al., 2013*; *Pugh et al., 2012*) that regulate unwinding activity, but there may be other possibilities. An intriguing example is the ARCH domain of XPD, which has been shown to interconvert dynamically between two conformations (*Ghoneim and Spies, 2014*). However, while these states are connected to DNA damage recognition (*Ghoneim and Spies, 2014*), their roles in unwinding have yet to be defined. The general mechanism described here for XPD may apply broadly to other helicases. Recent studies of SF2 helicase RecQ (*Harami et al., 2017*) suggest a similar inhibitory role of a secondary DNA binding site, which could plausibly be modulated by DNA binding proteins. Likewise, studies on SF1 helicases UvrD, Rep, and PcrA have shown that alternate conformations of one subdomain strongly affect activity, particularly processivity (*Comstock et al., 2015*; *Arslan et al., 2015*). Interactions with protein partners favoring particular conformations may provide a mechanism for activating processive unwinding (*Arslan et al., 2015*;

*Nguyen et al., 2017*). Identifying the molecular details of the conformational states that regulate helicase activity is a rich subject ripe for future investigation.

## Materials and methods

### Protein synthesis and purification

*Fac*XPD and *Fac*RPA2 were purified as described previously by *Pugh et al., 2008*. The H202A mutant was constructed by site-directed mutagenesis. A QuikChange Lightning site-directed mutagenesis kit (Agilent) was used to introduce an H202A substitution into the pET28a-FacXPD plasmid using CGGTTCAATGTTTTTGCTGGCACCGAAGGGG and CCCCTTCGGTGCCAGCAAAAACA TTGAACCG custom synthesized primers (Integrated DNA Technologies). Successful construction of the pET28a-FacXPD H202A plasmid was confirmed by sequencing at the DNA Core Sequencing Facility (University of Iowa IIHG Genomics Division). Mutant protein was expressed and purified identically to the wt XPD. T4 gp32 was obtained from a commercial source (New England Biolabs #M0300S).

### DNA construct synthesis

The hairpin construct was synthesized as described in *Qi et al., 2013*. The construct consists of three separate dsDNA fragments ligated together after synthesis and purification (*Figure 1—figure supplement 1*): a 1.5 kb 'right handle', an 89 bp 'hairpin' stem capped by a $(dT)_4$ loop, and a 1.5 kb 'left handle'. 'Right handle' and 'left handle' are functionalized with a 5′ digoxigenin and biotin, respectively, for binding to anti-digoxigenin and streptavidin-coated beads. The hairpin stem sequence used for all experiments was identical to 'sequence 1' used in *Qi et al., 2013*, which contains a random 49% GC sequence: 5′-GGC TGA TAG CTG AGC GGT CGG TAT TTC AAA AGT CAA CGT ACT GAT CAC GCT GGA TCC TAG AGT CAA CGT ACT GAT CAC GCT GGA TCC TA-3′. The hairpin stem is flanked by a 5′ 10 dT binding site for loading XPD and a 3′ abasic site to prevent XD unwinding into the handle. All oligonucleotides were purchased from Integrated DNA Technologies (Coralville, IA).

### Flow chamber for optical tweezers measurements

All measurements were carried out in laminar flow chambers described in *Whitley et al., 2017*. Chambers were made from NescoFilm (Karlan, Phoenix, AZ) melted between two glass coverslips and laser-engraved with channels for buffers. Chambers for all measurements had three channels (*Figure 1—figure supplement 2*). Top and bottom channels contained anti-digoxigenin and DNA-coated streptavidin beads, respectively. Small (100 µm OD) glass capillaries connected the top and bottom channels to the central channel for trapping and allow a controlled flow of beads. Three separate streams converged into the central trapping channel. Because the flow in each stream is laminar, mixing between different buffer streams is minimal and a reasonably sharp boundary between buffer conditions was maintained (*Figure 1—figure supplement 2a*, inset). This chamber design allowed moving freely between different streams via motorized sample stage and changing buffer conditions during an experiment, as described below.

### Optical trap measurements

Two high-resolution dual-trap optical tweezers instruments based on previously reported designs (*Whitley et al., 2017*; *Comstock et al., 2011*; *Bustamante et al., 2008*) were used to study XPD helicase unwinding in the presence of RPA2. The optical traps were calibrated by standard procedures (*Whitley et al., 2017*; *Berg-Sørensen and Flyvbjerg, 2004*) and both traps had a typical stiffness of $k = 0.3$ pN/nm in all experiments. All data were acquired using custom LabVIEW software (National Instruments, Austin, TX) available in an external repository (*Chemla, 2020a*; *Chemla, 2020b*). Most data were collected on one instrument at a 267 Hz sampling rate and boxcar filtered to a lower frequency, usually 89 Hz or as indicated in the text. Measurements with XPDH202A and gp32 were made on a second instrument at a 100 Hz sampling rate. All measurements were carried out at a constant force (ranging between 7 and 12 pN, as indicated in the text) using a feedback loop to maintain constant force.

Single XPD unwinding experiments were performed in a similar manner to that described in *Qi et al., 2013*, with some amendments. The trapping buffer consisted of 100 mM Tris-HCl (pH 7.6), 20 mM NaCl, 20 µM DTT, 3 mM MgCl$_2$, 0.1 mg/mL BSA, and oxygen scavenging system (*Swoboda et al., 2012*) (0.5 mg/mL pyranose oxidase [Sigma-Aldrich, St. Louis, MO], 0.1 mg/mL catalase [Sigma-Aldrich, St. Louis, MO], and 0.4% glucose) to increase the lifetime of the DNA tethers (*Landry et al., 2009*). To this buffer, varying concentrations of XPD, ATP, ATP-γS, and RPA2 or T4 gp32 were added.

During a typical experiment, the XPD–DNA complex was assembled by moving the traps between the three streams of the central trapping channel of the sample flow cell (*Figure 1—figure supplement 2a*). First, a single DNA hairpin was tethered between an optically trapped streptavidin-coated bead and an anti-digoxigenin-coated bead in the 'blank' stream, containing 500 µM ATP-γS but no protein or ATP (step 1 in *Figure 1—figure supplement 2b, c*). A low force (~2 pN) was applied to the tether, and the traps and tethered DNA were moved by motorized sample stage into the 'loading' stream containing 60 nM XPD + 500 µM ATP-γS, where they incubated for ~40–60 s to allow a single XPD to bind to the 10 dT ssDNA loading site but not unwind the hairpin (step 2). ATP-γS was used to increase the binding efficiency of XPD to the DNA loading site (*Honda et al., 2009*). Following incubation, the force on the tether was increased to a constant force (7–12 pN, as indicated) and the tether was moved into the 'unwinding' stream containing 500 µM ATP + 0–50 nM RPA2 or 250 nM T4 gp32 (step 3). (An exception to this protocol was used for testing XPD–RPA2 complex formation [*Figure 5—figure supplement 2*], in which case the loading stream contained 0–200 nM RPA2 in addition to XPD + ATP-γS, and the unwinding stream had no RPA2.) Upon exposure to ATP, a single XPD molecule bound at the loading site unwound the hairpin ahead of it, which, at constant force, resulted in an increase in the end-to-end extension of the DNA tether (*Figure 1—figure supplement 2c*). Unwinding data were typically collected until the tether broke or XPD dissociated.

For most measurements, the blank stream contained 10 nM of the dye molecule Cy3 in addition to 500 µM ATP-γS. Cy3 was added to this stream to detect, via fluorescence imaging, the precise locations of the stream boundaries (*Figure 1—figure supplement 2c*). Fluorescence detection of the Cy3 stream was achieved using a confocal microscope incorporated into the optical trap instrument, as described in *Whitley et al., 2017* and *Comstock et al., 2011*. Measurements with XPDH202A and gp32 were made on an instrument that did not have fluorescence capabilities, and thus Cy3 was not included.

## Analysis of DNA hairpins and determining base pairs unwound by XPD

A force–extension curve of each tether was usually taken in the middle stream containing no protein to verify proper synthesis. A properly synthesized hairpin unzips mechanically at an applied force of ~15 pN (*Figure 1—figure supplement 1b*), and its force–extension curve is well fit to the following model:

$$x(F) = N_{\mathrm{ds}}\xi_{\mathrm{ds}}(F) + N_{\mathrm{ss}}\xi_{\mathrm{ss}}(F), \tag{1}$$

where $\xi_{\mathrm{ds}}(F)$ and $\xi_{\mathrm{ss}}(F)$ are the extension of 1 bp of dsDNA and 1 nt of ssDNA at a given force, $F$, respectively, and $N_{\mathrm{ds}}$ and $N_{\mathrm{ss}}$ are the number of dsDNA base pairs and ssDNA nucleotides, respectively, in the construct. $\xi_{\mathrm{ds}}(F)$ and $\xi_{\mathrm{ss}}(F)$ are given by the extensible worm-like chain model of elasticity using the following parameters: persistence length $P_{ds} = 50$ nm and $P_{ss} = 1.0$ nm, contour length per base pair/nucleotide $h_{ds} = 0.34$ nm bp$^{-1}$ and $h_{ss} = 0.59$ nm nt$^{-1}$, and stretch modulus $S_{ds} = 1{,}000$ pN and $S_{ss} = 1{,}000$ pN (*Qi et al., 2013*; *Camunas-Soler et al., 2016*). For the closed hairpin at forces < 15 pN, the model *Equation (1)* was used with $N_{ds} = 3050$ bp, corresponding to the sum of the handle lengths, and $N_{ss} = 10$ nt, corresponding to the helicase loading site length (*Figure 1—figure supplement 1b*, black dashed line). For the open hairpin at forces > 15 pN, values $N_{ds} = 3050$ bp and $N_{ss} = 192$ nt were used (*Figure 1—figure supplement 1b*, gray dashed line).

As a helicase unwinds the tethered DNA hairpin at a constant force, the tether extension increases due to the release of 2 nt of ssDNA for each base pair of the hairpin dsDNA unwound. The number of base pairs unwound at each time point was obtained from the relation

$$n_{\text{unwound}}(t) = \frac{\Delta x(t)}{2\xi_{\text{ss}}(F)}, \qquad (2)$$

at the force $F$ applied, and where $\Delta x$ is the measured change in extension (in nm).

## Analysis of XPD unwinding bursts

Start and end times for each unwinding burst were selected manually using the following criteria: bursts were defined as periods of significant forward progress by XPD (>5 bp) followed by significant backward motion (>5 bp), excluding behavior obviously attributed to RPA2-mediated transient melting or 5 bp back- and forward step pairs XPD is known to take (*Qi et al., 2013*). The processivity (*Figure 2c*) for each burst was determined from the maximum number of base pairs open between the burst start and end time points, and the burst duration (*Figure 4—figure supplement 2*) was determined from the time elapsed between start and end time points.

For the analysis shown in *Figure 4*, we considered only the unwinding portion of each burst. For highly processive bursts, this segment consisted of the data from the beginning of the burst to the time point at which the maximum number of base pairs was unwound. However, for low-processivity bursts that exhibited extended stalls at ~10 bp, the unwinding endpoint was selected to be the last point above 10 bp. This allowed us to account for extended stalling behavior, while excluding rapid rezipping at the end of the burst. For bursts with processivity less than 10 bp, the unwinding portion of each burst was selected to be the first 75% of the total burst duration to exclude rapid rezipping at the end of the burst.

## Analysis of XPD unwinding speed

To calculate the unwinding speed during each burst, we first smoothed the data to remove any RPA2-mediated transient melting events. Data were smoothed over a 51-point (190 ms) window using the weighted local regression method 'rloess' in MATLAB's in-built 'smooth' function, which fits the span around each data point to a second-degree polynomial and excludes outliers based on a residual analysis. Then, the local velocity was determined over half-overlapping 50-point windows from the slope of a fit of the data to a line. The corresponding times and hairpin positions were also determined from their mean in each window. To create *Figure 4—figure supplement 1*, the local velocity for each burst was plotted against hairpin position and averaged over the set of bursts analyzed.

## Analysis of RPA2- and gp32-mediated melting events

RPA2- and gp32-like melting events on the bare hairpin were identified from short-lived melting of >4 bp of the hairpin (*Figure 3—figure supplement 2*). To find RPA2-like events during XPD unwinding, we identified steps in selected XPD unwinding bursts using an algorithm by *Kerssemakers et al., 2006* (*Figure 3—figure supplement 2a*). Specifically, we searched for adjacent pairs of steps that met the following criteria: >2 bp forward step (step $n$) followed by >2 bp back-step (step $n+1$) within five time points (56 ms) of each other (*Figure 3—figure supplement 2b* outlined in red). While false positives are observable even in the absence of RPA2, their probability is low, and the probability increases with RPA2 concentration (*Figure 3—figure supplement 2c*), confirming their connection to RPA2 activity. Each step was then aligned at full bandwidth (267 Hz) to the detected locations of RPA2 melting and reannealing. We used 2D kernel density estimation to visualize the probability density of these aligned traces. In this method, each point in the aligned data was replaced by a kernel function, here a 2D Gaussian. The sum of all kernels provides an excellent approximation to the probability density. The darker color in *Figure 3e, f* indicates higher probability.

## Kinetic model

We defined a model in which XPD can exist in two intrinsic states—a low- and high-processivity state—the latter of which we infer only when XPD unwinds past an ~25 bp threshold. In this model, XPD can alternate between 'low' and 'high' states (with rates constants $k_1$ and $k_{-1}$) and can also dissociate from DNA from both states (with rates constants $k_{off}$ and $k'_{off}$, respectively; *Figure 6b*). The effect of RPA2 increasing the probability of the high-processivity state is captured by the

dependence of the rate constant $k_1$ on [RPA2]. Since the high-processivity state is 'hidden', and unwinding bursts are scored as high processivity only when XPD unwinds >25 bp, the model includes the rate constant $k_2$ that determines the time XPD takes to pass this threshold.

We used this kinetic model to fit the measured first passage time for XPD to unwind past 25 bp. The model in **Figure 6b** is described by a set of master equations for the state probabilities:

$$
\begin{aligned}
\dot{P}_{low}(t) &= k_{-1}P_{high}(t) - \left(k_{off} + k_1\right)P_{low}(t) \\
\dot{P}_{high}(t) &= k_1 P_{low}(t) - \left(k'_{off} + k_{-1} + k_2\right)P_{high}(t) \\
\dot{P}_{off}(t) &= k_{off}P_{low}(t) + k'_{off}P_{high}(t) \\
\dot{P}_{>25}(t) &= k_2 P_{high}(t)
\end{aligned}
$$

where the subscripts correspond to the four possible states: low processivity ('low'), high processivity ('high'), dissociated ('off'), and past the 25 bp threshold ('>25'). The first passage time is measured from the time XPD is first exposed to RPA2, and XPD is in the low-processivity state. Thus, the initial conditions are given by

$$
P_{low}(0) = 1; P_{high}(0) = 0; P_{off}(0) = 0; P_{>25}(0) = 0
$$

These coupled linear differential equations can be solved to determine all four state probabilities. Here, we provide an expression for $P_{>25}$, which is plotted in **Figure 6d**:

$$
P_{>25}(t) = \frac{k_1 k_2}{\lambda_- - \lambda_+}\left(\frac{1 - e^{-\lambda_+ t}}{\lambda_+} - \frac{1 - e^{-\lambda_- t}}{\lambda_-}\right).
$$

The eigenvalues of the system give the decay constants for the exponentials:

$$
\lambda_\pm = \frac{k_1 + k_{-1} + k_2 + k_{off} + k'_{off}}{2} \pm \left[\left(\frac{k_1 + k_{-1} + k_2 + k_{off} + k'_{off}}{2}\right)^2 - \left(k_1\left(k_2 + k'_{off}\right) + k_{off}\left(k_{-1} + k'_{off}\right)\right)\right]^{1/2}.
$$

In our model, only $k_1$ is expected to vary as a function of RPA2 concentration. By grouping terms that are RPA2-concentration dependent vs. independent, $P_{>25}(t)$ can be written in terms of five unknown parameters $A$, $B$, $C$, $D$, and $k$ related to the five rate constants:

$$
P_{>25}(t, [\text{RPA2}]) = \frac{k([\text{RPA2}])D}{\lambda_- - \lambda_+}\left(\frac{1 - e^{-\lambda_+ t}}{\lambda_+} - \frac{1 - e^{-\lambda_- t}}{\lambda_-}\right), \tag{3}
$$

and

$$
\lambda_\pm = \frac{k([\text{RPA2}]) + B + C}{2} \pm \left[\left(\frac{k([\text{RPA2}]) + B + C}{2}\right)^2 - \left(Ak([\text{RPA2}]) + BC\right)\right]^{1/2}, \tag{4}
$$

where $k$ is written explicitly as a function of [RPA2]. The relationships between the five parameters and rate constants are given by

$$
k_1 = k([\text{RPA2}]); k_2 = D; k_{-1} = C - A; k'_{off} = A - D; k_{off} = B.
$$

We performed a global fit to our first-passage-time data (**Figure 6d**) using **Equations (3) and (4)** with common values $A$, $B$, $C$, and $D$ for all RPA2 concentrations but a different value for $k([\text{RPA2}])$ at each concentration (**Figure 6e**). In the data fit displayed in **Figure 6d** (dashed line), we set $k'_{off}$ to zero; we found that the best-fit value for $k'_{off}$ was small with a large error ($-0.05 \pm 1.23$) and provided no significant improvement to the fit. By assuming $k'_{off} = 0$, $A = D$, reducing our fit to four parameters: $A$, $B$, $C$, $k$. Parameter values and uncertainties are listed in **Table 2**.

In addition, we used **Equation (3)** to model the RPA2-dependent fraction of high-processivity bursts (**Figure 4b**) and average burst duration (**Figure 4—figure supplement 2**). For the former, data were fit to the expression for $P_{>25}(t)$ evaluated at a time $t$ equal to the average low-processivity burst duration $\tau_{low}$ 7 s. The latter was fit to the following model:

$$
\langle t_{burst} \rangle = \tau_{low}(1 - P_{>25}(t \to \infty, [RPA2])) + \tau_{high}P_{>25}(t \to \infty, [RPA2]),
$$

in which the average burst duration is given by the sum of the low-processivity burst duration, $\tau_{low}$, and the high-processivity burst duration, $\tau_{high}$, multiplied by the probabilities of being in their respective states at long times ($t \rightarrow \infty$):

$$P_{>25}(t \rightarrow \infty, [RPA2]) = \frac{k[\mathrm{RPA2}]D}{k[\mathrm{RPA2}]A + BC}.$$

## Acknowledgements

We thank the members of the Chemla and Spies laboratories for scientific discussions. Work in the Chemla lab was supported by the National Institutes of Health grant R01 GM120353. Work in the Spies lab was supported by the National Institutes of Health grant R35 GM131704.

## Additional information

### Competing interests
Maria Spies: Reviewing editor, *eLife*. The other authors declare that no competing interests exist.

### Funding

| Funder | Grant reference number | Author |
| --- | --- | --- |
| National Institutes of Health | R01 GM120353 | Yann R Chemla<br>Barbara Stekas<br>Steve Yeo<br>Alice Troitskaia |
| National Institutes of Health | R35 GM131704 | Maria Spies<br>Masayoshi Honda<br>Sei Sho |

The funders had no role in study design, data collection and interpretation, or the decision to submit the work for publication.

### Author contributions
Barbara Stekas, Conceptualization, Data curation, Formal analysis, Validation, Investigation, Writing - original draft, Writing - review and editing; Steve Yeo, Alice Troitskaia, Conceptualization, Data curation, Formal analysis, Validation, Investigation, Writing - review and editing; Masayoshi Honda, Sei Sho, Conceptualization, Resources; Maria Spies, Conceptualization, Resources, Writing - review and editing; Yann R Chemla, Conceptualization, Supervision, Funding acquisition, Investigation, Methodology, Writing - original draft, Writing - review and editing

### Author ORCIDs

Masayoshi Honda ⓘ http://orcid.org/0000-0001-8920-6301
Maria Spies ⓘ http://orcid.org/0000-0002-7375-8037
Yann R Chemla ⓘ https://orcid.org/0000-0001-9167-0234

### Decision letter and Author response
Decision letter https://doi.org/10.7554/eLife.60515.sa1
Author response https://doi.org/10.7554/eLife.60515.sa2

## Additional files

### Supplementary files
• Transparent reporting form

## Data availability

Summary data generated or analysed during this study are included in the manuscript and supporting files. Source data files have been provided for Figures 2, 3, 4, 5 and corresponding figure supplements.

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
