## [Decision Letter]

**Acceptance summary:**

The present manuscript explores the role of the ssDNA-binding protein RPA2 in regulating duplex DNA unwinding by XPD, an SF2 family helicase. Unwinding dynamics are studied using an optical tweezer assay, which reveals that RPA2 enhances XPD activity by increasing helicase processivity. The data are used to advance a model that RPA2 enhances processivity through a direct but transient interaction that activates a switch on XPD. Such a finding has the potential to have a strong influence on our understanding of helicase regulation by partner proteins.

**Decision letter after peer review:**

Thank you for submitting your article "Switch-like control of helicase processivity by single-stranded DNA binding protein" for consideration by *eLife*. Your article has been reviewed by two peer reviewers, and the evaluation has been overseen by a Reviewing Editor and Cynthia Wolberger as the Senior Editor. The reviewers have opted to remain anonymous.

The reviewers have discussed the reviews with one another and the Reviewing Editor has drafted this decision to help you prepare a revised submission.

Summary:

Using *Ferroplasma acidarmanus* proteins as a model system, the present manuscript explores the role of the ssDNA-binding protein RPA2 in regulating duplex DNA unwinding by XPD, an SF2 family helicase. Unwinding dynamics are studied using an optical tweezer assay, which reveals that RPA2 enhances XPD activity by increasing helicase processivity. The data are used to advance a model that RPA2 enhances processivity through a direct but transient interaction that activates a switch on XPD. Such a finding has the potential to have a strong influence on our understanding of helicase regulation by partner proteins.

The manuscript is well-written and presents high quality data that are well organized and easy to follow. The observation that increasing amounts of RPA2 leads to higher XPD processivity per se is well supported by the data. Unfortunately, the mechanism of processivity enhancement – that there is a physical RPA2/XPD contact – is not on similarly firm ground. Based on the results presented, alternative mechanisms still could be in play (some of which are that are outlined in the Introduction); in particular, the findings do not rule out the possibility that RPA2 may indirectly enhance processivity by helping to destabilize the ssDNA-dsDNA junction ahead of the helicase (see major comment 1). Moreover, the authors suggest in the Discussion that RPA2 binding at the ssDNA-dsDNA junction could prevent binding by a secondary site on XPD shown to regulate unwinding in related systems – this model fits very well to the stated observations, but was not directly tested. Overall, additional experiments should be performed to either support the proposed switch-like mechanism or to develop an alternative model.

Essential revisions:

1) In the Results section, "Melting of the hairpin by RPA2 does not aid XPD unwinding," the data do not exclude an indirect role for RPA2 in aiding XPD unwinding at the ssDNA-dsDNA junction. Very interesting bursts of melting are seen with increasing concentrations of RPA2, followed by rapid reannealing in the absence of XPD. These melting profiles, followed by reannealing features of the RPA2 bursts, are used as location fiducials in the presence of XPD. It is then examined whether XPD is more likely to melt after an RPA2 burst. Such an analysis assumes melting by RPA2 is always followed by reannealing before any possible enhancement by XPD. An alternative scenario is that RPA2-mediated melting is followed by rapid unwinding by XPD, which would result in no observable reannealing. The burst would thus look similar to normal XPD bursts but would exhibit high processivity since less melting would be required. Presumably, the melting by RPA2 is highly force-dependent. If the authors lower the force, they may be able to find a force regime where XPD-mediated unwinding is still stimulated, but where there are no transient melting events catalyzed by RPA2. If the RPA2 enhancement of unwinding is lost at the lower force, this would provide strong evidence that RPA2 melting occurs at the ssDNA-dsDNA junction to enhance helicase activity as opposed to through a direct interaction.

2) A similar processivity shift by XPD in the absence of RPA2 has been noted by the authors in a previous study. In that case, however, the change was triggered by XPD concentrations above 6 nM under constant flow conditions. This observation suggests a shift in processivity may a more general feature of XPD activity with multiple possible triggers beyond RPA2. The authors should determine whether other ssDNA binding proteins trigger the same shift to higher processivity.

3) In prior work, the authors argue that the higher processivity observed at XPD concentrations above 6 nM is the result of multiple helicases acting in concert. However, in the present manuscript, it is argued that only one helicase manifest higher processivity. How are these differing explanations reconcilable? The observation of processivity enhancement from the previous work should be discussed: one possible explanation is that additional XPDs shift the processivity by binding nearby each other in a manner similar to how RPA2 interfaces with XPD. Replacing RPA2 with a catalytically-dead form of XPD in the final flow lane would clarify whether the binding of additional helicases alone might trigger the enhancement. This experiment would also help address point 2.

4) The proposal that there exists a processivity "switch" which is activated by RPA2 is not well explained or supported by the results. Do the authors mean a switch-like conformational change in XPD? This has not been demonstrated in the manuscript. The results are consistent with several possible mechanisms that are not all switch-like. The model should be clarified and it should be discussed how it is supported by current observations.

5) In the Discussion, the authors suggest RPA2 binding at the ssDNA-dsDNA junction could prevent binding by a secondary site on XPD shown to regulate unwinding in related systems. This model fits very well to the observations made, but was not directly tested. The authors should consider introducing mutations that disrupt the activity of the secondary site to determine whether there is an enhancement of helicase unwinding similar to that observed with RPA2.

6) The authors provide a kinetic model to describe the two states of XPD (low processivity and high processivity). The model provides a good fit to the fraction of molecules in the high processivity mode at different concentrations of RPA2. The authors mention a second model that included an off-rate for the high processivity state. Based on the simple idea that kinetic descriptions of processivity typically include an off-rate for each identified species, it seems that a model which includes this feature (an off-rate) for the high processivity species would be preferred. If possible, the authors could provide additional explanation, perhaps in the supplement, as to why they favor the model depicted in Figure 5 over a model which includes an explicit rate constant for dissociation of the high processivity species.

---

## [Author Response]

Essential revisions:1) In the Results section, "Melting of the hairpin by RPA2 does not aid XPD unwinding," the data do not exclude an indirect role for RPA2 in aiding XPD unwinding at the ssDNA-dsDNA junction. Very interesting bursts of melting are seen with increasing concentrations of RPA2, followed by rapid reannealing in the absence of XPD. These melting profiles, followed by reannealing features of the RPA2 bursts, are used as location fiducials in the presence of XPD. It is then examined whether XPD is more likely to melt after an RPA2 burst. Such an analysis assumes melting by RPA2 is always followed by reannealing before any possible enhancement by XPD. An alternative scenario is that RPA2-mediated melting is followed by rapid unwinding by XPD, which would result in no observable reannealing. The burst would thus look similar to normal XPD bursts but would exhibit high processivity since less melting would be required.

If what the reviewers propose is true, then we would expect XPD unwind traces in the presence of RPA2 to consist of +5 bp upward steps due to RPA2 melting, followed by a plateaus during which XPD rapidly translocates to catch up to the fork position, and then +1 bp steps as XPD takes the lead unwinding at the fork junction. However, visual inspection of individual unwinding traces show that such events are rarely observed. Analyzing the occurrence of +5 bp / +1 bp step pairs shows that these events actually become 2 times less frequent as RPA2 concentration increases, in contrast to +5 bp / -5 bp pairs (Figure 3C).

We have included this analysis in the revised Results section. However, we believe we have more compelling evidence against RPA2 melting as the mechanism of enhancement. This evidence is described in more detail below and in our response to Essential revision #2.

Presumably, the melting by RPA2 is highly force-dependent. If the authors lower the force, they may be able to find a force regime where XPD-mediated unwinding is still stimulated, but where there are no transient melting events catalyzed by RPA2. If the RPA2 enhancement of unwinding is lost at the lower force, this would provide strong evidence that RPA2 melting occurs at the ssDNA-dsDNA junction to enhance helicase activity as opposed to through a direct interaction.

We thank the reviewers for this excellent suggestion. In the revised manuscript we have collected additional data at different forces to test the RPA2 melting mechanism. As expected, we find that transient melting by RPA2 decreases at lower forces and is almost undetectable for forces <9 pN at the RPA2 concentrations assayed (see revised Figure 3—figure supplement 3). XPD processivity shows a weak dependence on force over the range 7-12 pN, consistent with our prior studies (Qi et al., 2013). Thus, as suggested by the reviewers, we identified a force around 9 pN where RPA2 melting is lost but XPD-mediated activity is still observed. At this force, we still find a significant enhancement in XPD processivity due to the presence of RPA2 (see revised Figure 2—figure supplement 1).

This result, along with additional measurements (see Essential Point #2), indicate that RPA2-mediated melting is not required to enhance XPD processivity. We present these new data in the revised Results section.

2) A similar processivity shift by XPD in the absence of RPA2 has been noted by the authors in a previous study. In that case, however, the change was triggered by XPD concentrations above 6 nM under constant flow conditions. This observation suggests a shift in processivity may a more general feature of XPD activity with multiple possible triggers beyond RPA2. The authors should determine whether other ssDNA binding proteins trigger the same shift to higher processivity.

The reviewers raise an important point regarding the effect of XPD concentration on processivity, which we address in Essential revision #3 below. In our response here, we focus on whether other ssDNA binding proteins can enhance XPD processivity similarly to RPA2.

In previous ensemble studies (Pugh et al., 2008), we probed the effect of heterologous ssDNA-binding proteins (including *E. coli* SSB, T4 gp32, and two archaeal RPAs) on XPD activity, and showed that the strongest enhancement occurred with RPA2 (see Figure 3 in Pugh et al.). These results provide strong evidence that the effect of RPA2 on XPD unwinding activity is specific. To investigate this question further, we carried out additional single-molecule experiments with the single-stranded DNA binding protein gp32 from bacteriophage T4. We selected gp32 because, like RPA2, it is a small protein consisting of a single OB fold and has a similar footprint on ssDNA. Furthermore, we identified conditions (force and gp32 concentration) in which gp32 transiently melts DNA by a similar amount (5-10 bp) and at a similar frequency (~4%) as RPA2 over the concentration of range assayed. However, we observe no effect of gp32 on XPD processivity (see revised Figure 3—figure supplement 4).

Along with our data at lower force (Essential Point #1), these new measurements with a non-cognate protein prove that transient melting of the duplex is insufficient, on its own, to enhance XPD activity and support our model that specific XPD-RPA2 interactions are required. We present these new findings in the revised Results section.

3) In prior work, the authors argue that the higher processivity observed at XPD concentrations above 6 nM is the result of multiple helicases acting in concert. However, in the present manuscript, it is argued that only one helicase manifest higher processivity. How are these differing explanations reconcilable? The observation of processivity enhancement from the previous work should be discussed: one possible explanation is that additional XPDs shift the processivity by binding nearby each other in a manner similar to how RPA2 interfaces with XPD. Replacing RPA2 with a catalytically-dead form of XPD in the final flow lane would clarify whether the binding of additional helicases alone might trigger the enhancement. This experiment would also help address point 2.

We believe the reviewers may have missed an important aspect of the single-molecule experiments in the current manuscript: all the measurements were made in such a way that *only a single XPD* could unwind. In our sample flow cell (Figure 1—figure supplement 2), the DNA hairpin tethered to trapped beads is first placed in a buffer stream containing XPD but no hydrolyzable ATP to allow the helicase to bind to DNA but *not unwind* it. Since the hairpin construct contains a ssDNA binding site equal in length to XPD’s footprint (10 nt), it allows only a single XPD to bind. Then, the trapped hairpin and bound XPD are moved inside the flow cell to a stream containing ATP (and RPA2) but no XPD. These measures ensure that only a single XPD unwinds at a time and that no secondary XPD’s can bind from solution during unwinding. Once the single XPD dissociates from the hairpin, we observe all unwinding activity to cease, confirming the absence of other XPD molecules in solution. Note that under this experimental protocol, the concentration of XPD has no impact on the detected unwinding activity since the activity always results from a single molecule.

In our prior work (Qi et al., 2013) referenced by the reviewers, some measurements were made to determine the effect of multiple XPDs on unwinding activity. This was achieved either by extending the ssDNA binding site (>10 nt) to accommodate multiple XPDs (see Figure 2A-E in Qi et al.) or by using a sample flow cell layout in which both XPD and ATP were in the same stream, so that multiple XPD molecules could bind and unwind the DNA (see Figure 2F-H). Only in the latter approach is the 6 nM XPD concentration relevant to the measured processivity, since it determines the XPD occupancy on the DNA.

We have revised the Results section, the Materials and methods section, and the figures (see Figure 1—figure supplement 2) to make this point clearer.

4) The proposal that there exists a processivity "switch" which is activated by RPA2 is not well explained or supported by the results. Do the authors mean a switch-like conformational change in XPD? This has not been demonstrated in the manuscript. The results are consistent with several possible mechanisms that are not all switch-like. The model should be clarified and it should be discussed how it is supported by current observations.

We address this question in our response to Essential revision #5 below.

5) In the Discussion, the authors suggest RPA2 binding at the ssDNA-dsDNA junction could prevent binding by a secondary site on XPD shown to regulate unwinding in related systems. This model fits very well to the observations made, but was not directly tested. The authors should consider introducing mutations that disrupt the activity of the secondary site to determine whether there is an enhancement of helicase unwinding similar to that observed with RPA2.

We thank the reviewers for this suggestion. Based on our previous mutagenesis work (Pugh et al., 2012), we introduced a point mutation, H202A, in a known secondary site for the translocating DNA strand that regulates XPD force generation. This mutation resides in Helicase Domain 1 (HD1) of XPD and affects the secondary DNA binding site located in the groove between HD1 and the FeS domains, which directs the translocating strand out of the pore between the HD1 and ARCH domains. H202A is analogous to the H198A mutation in the related *T. acidophilum* XPD helicase, which Pugh at al. determined to have the strongest positive effect on helicase activity. Our new single-molecule measurements (see revised Figure 5 and Figure 5—figure supplement 1) show that disrupting the secondary DNA binding site with the H202A mutation leads to a dramatic increase in unwinding processivity compared to the wild-type protein, at two different forces. Furthermore, we find that RPA2 does not enhance XPDH202A unwinding further, and that the processivity of XPDH202A is similar to that of wild-type XPD in the presence of saturating RPA2. Our new results show that interfering with this regulatory DNA binding site by mutation elicits the same effect on XPD as RPA2. We present these new findings in the revised Results section.

The data collected for the revised manuscript support a model in which RPA2 interacts directly with the XPD-DNA complex and interferes with DNA contacts made to this secondary site. Current understanding is that this site helps position the DNA fork relative to the pore through which the strands are separated (Pugh et al., 2012). We propose that different fork conformations modulate XPD unwinding activity and force generation, and that RPA2 (similarly to a H202 mutation) acts by stabilizing conformation(s) that are competent for high-processivity unwinding. This switch-like behavior is manifested in the two types of unwinding bursts observed. This model is presented in the extensively revised Discussion section and in Figure 6.

6) The authors provide a kinetic model to describe the two states of XPD (low processivity and high processivity). The model provides a good fit to the fraction of molecules in the high processivity mode at different concentrations of RPA2. The authors mention a second model that included an off-rate for the high processivity state. Based on the simple idea that kinetic descriptions of processivity typically include an off-rate for each identified species, it seems that a model which includes this feature (an off-rate) for the high processivity species would be preferred. If possible, the authors could provide additional explanation, perhaps in the supplement, as to why they favor the model depicted in Figure 5 over a model which includes an explicit rate constant for dissociation of the high processivity species.

As explained in the Materials and methods section, a model including dissociation from the high-processivity state was first considered. However, the best fit values returned a low value for the corresponding rate constant *k_off_’* with a large error, which signified that the dissociation rate is negligible compared to that from the low-processivity state (*k_off_*) and poorly determined from fitting the data. Setting this parameter to 0 had no discernible effect on the fit quality.